# Targeting RNA structure in *SMN2* reverses spinal muscular atrophy molecular phenotypes

Amparo Garcia-Lopez[1], Francesca Tessaro[1], Hendrik R.A. Jonker[2], Anna Wacker[2], Christian Richter[2], Arnaud Comte[3], Nikolaos Berntenis[4], Roland Schmucki[4], Klas Hatje[4], Olivier Petermann[1], Gianpaolo Chiriano[1], Remo Perozzo[1], Daniel Sciarra[1], Piotr Konieczny[5,6], Ignacio Faustino[7], Guy Fournet[3], Modesto Orozco[7], Ruben Artero [5,6], Friedrich Metzger[4], Martin Ebeling[4], Peter Goekjian[3], Benoît Joseph[3], Harald Schwalbe[2] & Leonardo Scapozza[1]

Modification of *SMN2* exon 7 (E7) splicing is a validated therapeutic strategy against spinal muscular atrophy (SMA). However, a target-based approach to identify small-molecule E7 splicing modifiers has not been attempted, which could reveal novel therapies with improved mechanistic insight. Here, we chose as a target the stem-loop RNA structure TSL2, which overlaps with the 5′ splicing site of E7. A small-molecule TSL2-binding compound, homocarbonyltopsentin (PK4C9), was identified that increases E7 splicing to therapeutic levels and rescues downstream molecular alterations in SMA cells. High-resolution NMR combined with molecular modelling revealed that PK4C9 binds to pentaloop conformations of TSL2 and promotes a shift to triloop conformations that display enhanced E7 splicing. Collectively, our study validates TSL2 as a target for small-molecule drug discovery in SMA, identifies a novel mechanism of action for an E7 splicing modifier, and sets a precedent for other splicing-mediated diseases where RNA structure could be similarly targeted.

[1] Pharmaceutical Biochemistry Group, School of Pharmaceutical Sciences, University of Lausanne and University of Geneva, Rue Michel-Servet 1, 1211 Geneva, Switzerland. [2] Institut für Organische Chemie und Chemische Biologie, Center for Biomolecular Magnetic Resonance (BMRZ), Johann Wolfgang Goethe-University Frankfurt, Max-von-Laue-Strasse 7, 60438 Frankfurt, Germany. [3] Institut de Chimie et Biochimie Moléculaires et Supramoléculaires (ICBMS), UMR CNRS 5246, University Claude Bernard Lyon 1, 43 Bd du 11 Novembre 1918, F-69622 Villeurbanne cedex, France. [4] Pharmaceutical Research and Early Development, F. Hoffmann-La Roche, Roche Innovation Center Basel, Grenzacherstrasse 124, 4070 Basel, Switzerland. [5] Translational Genomics Group, Incliva Health Research Institute, Menendez Pelayo 4, 46010 Valencia, Spain. [6] Department of Genetics and Interdisciplinary Research Structure for Biotechnology and Biomedicine (ERI BIOTECMED), University of Valencia, Dr Moliner 50, 46100 Burjassot, Spain. [7] Institute for Research in Biomedicine (IRB), Barcelona Institute of Science and Technology (BIST), Joint BSC-IRB Research Program in Computational Biology, Baldiri Reixac 10, 08028 Barcelona, Spain. These authors contributed equally: Francesca Tessaro, Hendrik R.A. Jonker, Anna Wacker. Correspondence and requests for materials should be addressed to A.G.-L. (email: amparo.garcialopez@unige.ch) or to L.S. (email: leonardo.scapozza@unige.ch)

Spinal muscular atrophy (SMA, OMIM #253300) is an autosomal recessive disorder that causes degeneration of α-motor neurons in the spinal cord. There is at present no cure or effective therapy for this disease, which is the most common genetic cause of infant mortality. SMA is primarily caused by deletions or homozygous mutations in the survival motor neuron 1 (SMN1) gene[1]. This gene encodes a ubiquitous 38-kDa protein (SMN) present in the cytoplasm and the nucleus, with a direct role in mRNA metabolism[2, 3]. From a clinical point of view, five types of SMA (0, I, II, III and IV) have been described, ranging from complete absence of motor function and infant mortality, to minor motor defects with no significant reduction in lifespan. The severity of symptoms in SMA strongly correlates with the levels of SMN protein, with 30% variations being sufficient to transition from mild to severe forms of the disease[4].

A second copy of the SMN1 gene called SMN2 is present in the human genome. However, SMN2 has never been found mutated in SMA patients. SMN1 and SMN2 are nearly identical. The only difference in their coding region is a translationally silent C>T transition at position +6 of exon 7 (E7) in SMN2[5], which results in different alternative splicing patterns. In SMN1, E7 is included giving a full-length SMN transcript that encodes a functional SMN protein. However, the C>T transition in SMN2 causes E7 exclusion in the majority of cases, producing a C-terminally truncated SMN protein that is not active. Only ~10% of the SMN2 mRNAs include E7 and give full-length SMN protein[6].

Promoting SMN2 E7 inclusion to increase the amount of full-length SMN that this gene can generate has been demonstrated to functionally compensate for the lack of SMN1 in vivo[7–9], validating the therapeutic potential of targeting SMN2 splicing. Some approaches employed to date to modify SMN2 splicing include antisense oligonucleotides (ASOs) and a variety of small molecules identified through splicing-reporter phenotypic screenings[8–14]. However, despite the current good understanding on the cis and trans-acting factors that regulate SMN2 E7 splicing[15], target-based approaches to screen for SMN2 splicing modifier compounds have not yet been attempted. Such target-based strategies have the potential to identify new chemical entities for SMA, the development of which would be facilitated by their associated mechanistic insight. Among the cis-regulatory elements that regulate E7 inclusion, a 19-nt RNA hairpin known as TSL2 (terminal stem-loop 2) located at the exon 7/intron 7 junction of the SMN transcripts plays a key role (Supplementary Fig. 1)[16, 17]. It was previously proposed that the 3′ end of the TSL2 hairpin partially sequesters the 5′ splice site (5′ ss) of E7, negatively affecting the recruitment of the splicing machinery and exon inclusion[16, 17]. Consistently, point mutations in SMN2 that disrupt TSL2 promote E7 splicing to levels that in some cases match SMN1[16].

Despite recent efforts[18, 19], RNA remains an underexploited target for small-molecule drug discovery, mainly due to the limited information available on RNA–ligand interactions and the poor suitability of current high-throughput screening assays for RNA[20]. However, during the last decade, it has become apparent that RNA structure determines the outcome of mRNA processing in a growing number of examples, and is now recognised as a key player in many human diseases[21]. In this study, we perform a target-based small-molecule screening for SMA using the TSL2 RNA structure as the biological target, and we identify a hit compound that increases SMN2 E7 splicing by stabilising a conformation of TSL2 that improves accessibility of the 5′ splice site. These findings open new avenues for drug discovery in SMA, as well as for other splicing-mediated diseases where functional RNA structures could be similarly targeted.

## Results

**TSL2 is a suitable target for small-molecule screening.** Success of target-based screening strongly relies on the adequate selection of a biological target. Therefore, we first sought to validate the suitability of TSL2 as a target for small-molecule screening. To this end, HeLa cells were transiently transfected with a battery of SMN1 and SMN2 minigenes containing exons 6-to-8 and bearing different structural mutations in TSL2. These included mismatch mutations (3C, 4G and 6C), a mutation in the loop (9C) and a base-pairing strengthening mutation (2C). All mutations were located in the 5′ half of the TSL2 hairpin, in order to avoid interference with recognition of the 5′ ss of E7 at the sequence level (Fig. 1a). SMN2 minigenes carrying mismatch mutations in the TSL2 stem (3C, 4G and 6C), as well as the loop mutation (9C), showed increased SMN2 E7 inclusion levels, which ranged from 16% (no mutation) to 62% (mutation 6C) (Fig. 1b, c). Conversely, strengthening of base pairing in TSL2 (mutation 2C) promoted E7 skipping in SMN1. These results are in agreement with a previous report using a different cell line[16].

To confirm that the effect of these mutations on SMN2 splicing was linked to conformational changes in TSL2, a series of low-resolution structural methods were used on synthetic TSL2 RNAs, both in their mutated (2C, 3C, 4G, 6C and 9C) and non-mutated versions (Supplementary Table 1). First, TSL2 base stacking was indirectly measured by labelling the TSL2 RNAs with the fluorescent structural probe 2-aminopurine (2AP)[22], the emission of which is quenched by RNA base contacts. 2AP fluorescence measurements showed that all mutations induced RNA-stacking changes that significantly correlated with E7 inclusion, with the exception of the 4G mutation (Fig. 1d). Visualisation of hairpin formation using native PAGE confirmed these findings (Fig. 1e and Supplementary Fig. 2). The 4G mutation triggered the strongest conformational changes in TSL2. However, SMN2 splicing was only mildly affected, suggesting that a certain level of TSL2 structure is required for exon inclusion, or that a protein-binding sequence may have been affected by this mutation.

Circular dichroism (CD) spectroscopy was used to further confirm these observations using non-labelled TSL2 RNAs. In these experiments, all TSL2 RNAs showed a positive CD peak at ~265 nm typical of RNA and a negative peak at ~210 nm typical of the A-form. At low temperature (10 °C), all mutations caused either a reduction in the CD signal of TSL2 (4G, 9C; Fig. 1f) or a significant wavelength shift in the positive peak, from 265 to 269 nm (3C, 6C; Fig. 1f), indicative of changes in base stacking that were consistent with our 2AP and PAGE results (Fig. 1d, e). At a denaturing temperature (90 °C), all differences in the CD spectra disappeared (Supplementary Fig. 3), demonstrating that the changes triggered by these mutations represent conformational features rather than sequence effects.

In summary, these results demonstrate that triggering conformational changes in TSL2 can increase SMN2 E7 splicing values to nearly SMN1 levels, and confirm the potential of TSL2 as a target for the screening of RNA structure-modifying small molecules able to induce equivalent changes.

**A target-based screening identifies TSL2-binding molecules.** Screening efforts to find RNA-binding compounds typically yield low hit rates. However, hit rates are reported to increase from ~1% to ~19% when using small molecules with chemical scaffolds biased for RNA recognition; including indole, 2-phenyl indole, 2-phenyl benzimidazole and alkyl pyridinium (Fig. 2a)[23]. Based on this information, we conducted in silico filtering of an in-house, high-chemical diversity structure database of ~3000 compounds for the creation of a focused library of 304 small molecules

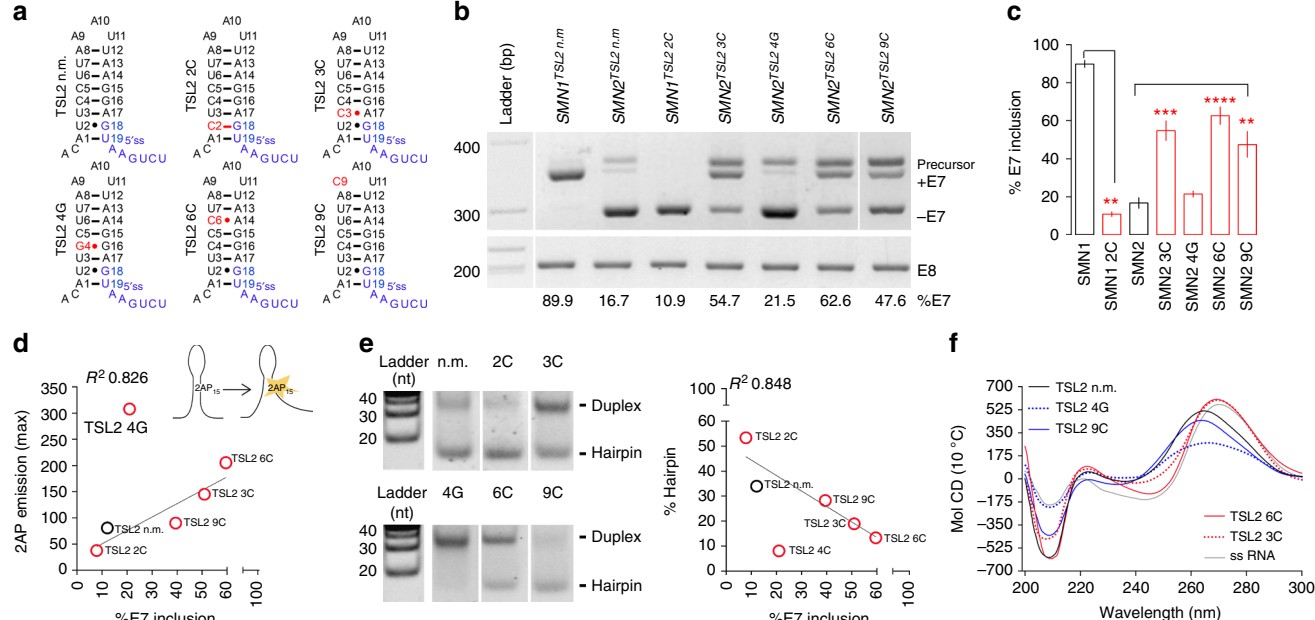

**Fig. 1** Structural mutations in TSL2 affect *SMN2* exon 7 inclusion. **a** Predicted secondary structure of non-mutated (n.m.) TSL2 or TSL2 carrying the indicated mutations (red). The 5′ splice site (5′ ss) of exon 7 (E7) is shown (blue). **b**, **c** Agarose gel image showing RT-PCR products from HeLa cells transfected with $SMN^{E6-to-8}$ minigenes[16] carrying either n.m. TSL2 or mutated TSL2. Mutations that disrupt the TSL2 stem (3C, 4G and 6C) and a mutation in the loop (9C) improve *SMN2* E7 splicing. A mutation that strengthens the TSL2 stem (2C) prevents *SMN1* E7 inclusion (*n* = 9; three biological and three technical replicates). Exon 8 (E8) was unaffected. A higher molecular-weight band corresponding to precursor *SMN* mRNA is detected[16]. For simplicity, this band is omitted in subsequent gel images. **d** Correlation between E7 inclusion and TSL2 base-stacking changes caused by TSL2 mutations, measured as the increase in fluorescence of synthetic 2-aminopurine (2AP)-labelled TSL2 RNAs (*n* = 8; two plates with four replicates each). 2AP was introduced at position 15, substituting its analogue G. **e** Correlation between E7 inclusion and in vitro hairpin formation, measured from native polyacrylamide gels of synthetic TSL2 RNAs folded by snap cooling (*n* = 3 independent gels). The RNA ladder is single stranded. Mutations that increase E7 splicing reduce TSL2 hairpin formation in vitro and favour duplex interactions. **f** Circular dichroism (CD) spectra (10 °C) of TSL2 RNAs (*n* = 10 scans). Reduced negative-peak intensities (~210 nm) indicate reduced helicity. Reduced positive signals (~265 nm) indicate reduced stacking. Wavelength shifts suggest additional rearrangements. ss RNA single-stranded RNA (control), Mol CD molar circular dichroism units. *$p < 0.5$, **$p < 0.01$ and ***$p < 0.001$. The graph in **c** represents mean values ± SEM. *p*-values were obtained by applying non-paired, two-tailed *t* tests with Welch corrections

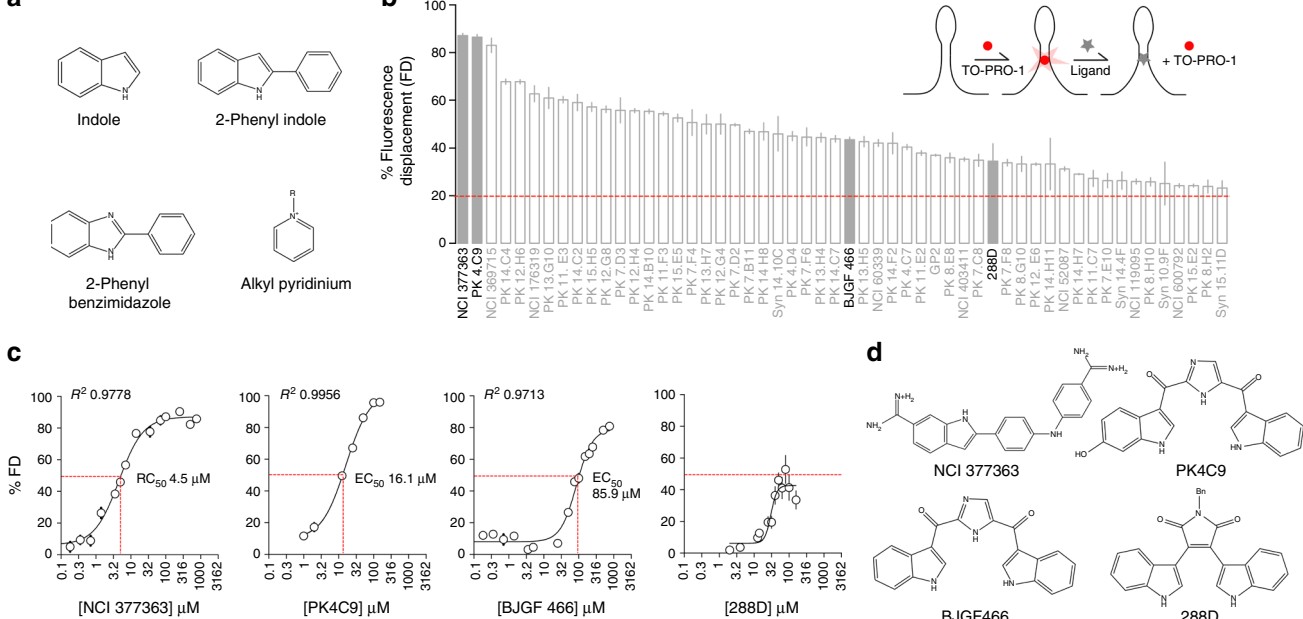

**Fig. 2** In vitro screening of a focused library identifies TSL2-interacting small molecules. **a** Selected scaffolds for the building of a 304-compound, focused RNA-binding chemical library. **b** Ranking of positive hits (100 μM) from the primary TSL2-interacting screening, with a cut-off value of 20% fluorescence displacement (FD) relative to DMSO (1%) controls (*n* = 4). Hits NCI377363, PK4C9, BJGF466 and 288D are highlighted. **c** EC50 binding curves and values of hits NCI377363, PK4C9 and BJGF466 to TSL2 (*n* = 8; two plates with four replicates each). An EC50 value for 288D could not be calculated due to poor solubility. **d** Chemical structures of NCI377363, PK4C9, BJGF466 and 288D. Graphs represent mean values ± SEM

containing such RNA-binding scaffolds. This collection was tested in 96-well-plate format (100 μM) against a synthetic TSL2 RNA (0.5 μM) using a fluorescence displacement (FD) assay, with the fluorescent dye TO-PRO-1 as our probe[24]. The Z-factor of this assay was 0.7, confirming the suitability of the screening method[25].

The primary FD screening led to the identification of 54 TSL2-interacting hits (hit rate 17%, Fig. 2b and Supplementary Data 1), with EC50 values starting at 4.5 μM (Fig. 2c). Among all hits, 45 molecules showed good dose responses (Supplementary Fig. 4), and were subsequently tested in our 2AP assay to confirm their ability to induce conformational changes in TSL2. A total of 14 out of 28 hits showed a significant effect on TSL2 conformation, with seven hits increasing RNA base stacking and seven reducing it (Supplementary Fig. 5). A total of 17 molecules could not be tested due to autofluorescence interfering with the emission of 2AP. An unrelated RNA structure and a denatured TSL2 RNA were used to assess the binding selectivity of our four most promising hits (Fig. 2d), with three of them showing good binding discrimination (Supplementary Fig. 6).

These results represent the first example of TSL2 structure-modifying small molecules, as well as the first target-based screening reported for SMA.

**TSL2 binders modify *SMN2* E7 splicing in different systems**. To assess the activity on *SMN2* E7 splicing of the identified hits, 19 candidates were selected for a secondary screen in HeLa cells transfected with the *SMN2* minigene. RT-PCR identified eight molecules (42% of total tested) that significantly increased E7 inclusion after 6, 12 and 24 h of treatment, relative to control cells treated with 0.04% DMSO (Fig. 2d; Fig. 3a, b and Supplementary Table 2). Of these, marine natural molecule homo-carbonyltopsentin (PK4C9) showed the strongest effect, with an average E7 inclusion of up to 72% at 40 μM (43% increase with respect to DMSO-treated cells) and an EC50 value of ~25 μM, consistent with its EC50 value in the TO-PRO-1 binding assay (16 μM, Fig. 2c). Importantly, this effect depended on the integrity of TSL2, as the use of minigenes carrying structural mutations in TSL2 affected PK4C9 activity (see section "PK4C9 improves accessibility of the 5′ ss of *SMN2* E7" below). Moreover, eight in-house-produced analogues of PK4C9 with reduced TSL2 binding ability also displayed decreased activity on E7 inclusion, further demonstrating that the splicing modifier activity of PK4C9 is mediated by its interaction with TSL2 (Supplementary Fig. 7).

To study the effect of our screening hits on endogenous *SMN2* splicing in a pathological context, six of them were subsequently tested on a fibroblast cell line derived from a type-I SMA patient (GM03813C). All six molecules showed an increase of E7 splicing that was consistent with our results in HeLa (Fig. 3b), with PK4C9 having the strongest effect and the lowest cytotoxicity (Fig. 3d and Supplementary Fig. 8). Upon 24-h treatment with PK4C9 at 40 μM, E7 inclusion increased up to 97% (41% increase with respect to DMSO-treated cells). Phenotypically unaffected fibroblasts carrying a mutated copy of *SMN1* in heterozygosis (GM03814B) were also used as an additional cell line. Here, E7 inclusion reached a maximum of 91% upon 24-h treatment with PK4C9 at 40 μM (25% increase with respect to DMSO-treated cells) (Fig. 3b). qRT-PCR from molecule-treated vs. DMSO-treated GM03813C cells confirmed these results, showing an up to 5.2-fold decrease in the expression of E7-excluding *SMN2* isoforms, and up to three-fold increase in E7-including isoforms (Fig. 3c).

Finally, PK4C9 was tested on transgenic *Drosophila* flies as an in vivo reporter of *SMN2* splicing in motor neurons. In these flies,

the expression of a *UAS-SMN2* transgene carrying human *SMN2* exons 6-to-8[26] (*SMN2$^{E6-to-8}$*) was targeted to motor neurons using the *D24-Gal4* driver line (25 °C). The splicing pattern of *SMN2$^{E6-to-8}$* in *Drosophila* promotes E7 inclusion to a higher extent than human cells (72% inclusion). Upon oral treatment with PK4C9 at 200 μM, this percentage increased to 84% (Fig. 3e), hence validating the effect of PK4C9 in an additional cell type and in the context of a whole organism with a highly conserved blood–brain barrier[27, 28].

Collectively, our results demonstrate that screening for TSL2-interacting molecules can effectively identify splicing modifiers of *SMN2* that are active in different model systems, including HeLa, SMA fibroblasts and *Drosophila* motor neurons, thus validating our target choice and screening strategy.

**PK4C9 rescues functional SMN protein in SMA cells**. To correlate the PK4C9-induced effect on *SMN2* E7 splicing with an increase in full-length SMN protein, total levels of SMN were measured by western blot in GM03813C and GM03814B fibroblasts. Before treatment, phenotypically unaffected GM03814B cells showed 2.3 times more SMN protein than GM03813C SMA fibroblasts (Fig. 4a). However, upon treatment of GM03813C cells with PK4C9 (40 μM, 48 h), SMN protein increased 1.5-fold compared to GM03813C cells treated with DMSO (Fig. 4a). This increase is in agreement with values reported for other E7 splicing modifier small molecules[8, 9].

The effect of PK4C9 on SMN subcellular localisation was also studied in GM03813C and GM03814B fibroblasts by immuno-histochemistry. Wild-type distribution of SMN includes both the nucleus and the cytoplasm. In the nucleus, functional SMN oligomerises and forms punctate aggregates known as gems, which co-localise with coilin-p80[2]. To confirm the production of SMN protein with restored oligomerisation capacity, double-antibody staining of coilin-p80 and SMN was used to compare the number of gems in PK4C9-treated cells (40 μM) vs. DMSO-treated controls (24 h). A significant increase in the number of SMN-positive gems and coilin-p80-positive gems per cell was detected in both lines compared to DMSO-treated cells. Moreover, the percentage of cells with gems was also elevated in PK4C9-treated cells (Fig. 4b, c and Supplementary Fig. 9).

SMN loss-of-function has been linked to widespread pre-mRNA splicing defects[29−32]. To further confirm the functionality of the restored SMN protein, we next studied whether PK4C9 could rescue SMN-loss-mediated splicing defects using RNA-sequencing (RNA-seq). Statistical analysis revealed 290 transcripts with modified splicing in PK4C9-treated (40 μM, 24 h) GM03813C fibroblasts compared to DMSO controls (delta percent spliced-in index PSI > 0.4), corresponding to 201 single genes, including *SMN2* (Supplementary Data 2 and Supplementary Table 3). Gene set enrichment analysis (GSEA) did not return significantly over-represented biological pathways, indicating that these splicing changes correspond to a mechanistic effect rather than a concerted toxic response. To discern whether these alterations represented off-targets or a direct consequence of SMN recovery, qRT-PCR was performed on PK4C9-treated vs. DMSO-treated wild type (WT, ND36091A) and SMA fibroblasts for eight of the 201 genes. We expected that (1) SMN-dependent changes would respond differently to treatment in WT vs. SMA cells, given their different SMN starting levels; whereas (2) true off-targets would be similarly affected. Four of these eight genes (*RPS6KB1*, *LPIN1*, *PHF14* and *INSR*) showed PK4C9 dose-sensitive responses in SMA fibroblasts but not in WT cells, confirming that at least part of the PK4C9-induced splicing changes are a consequence of SMN recovery (Fig. 4d; Supplementary Fig. 10 and Supplementary Data 3). Consistently,

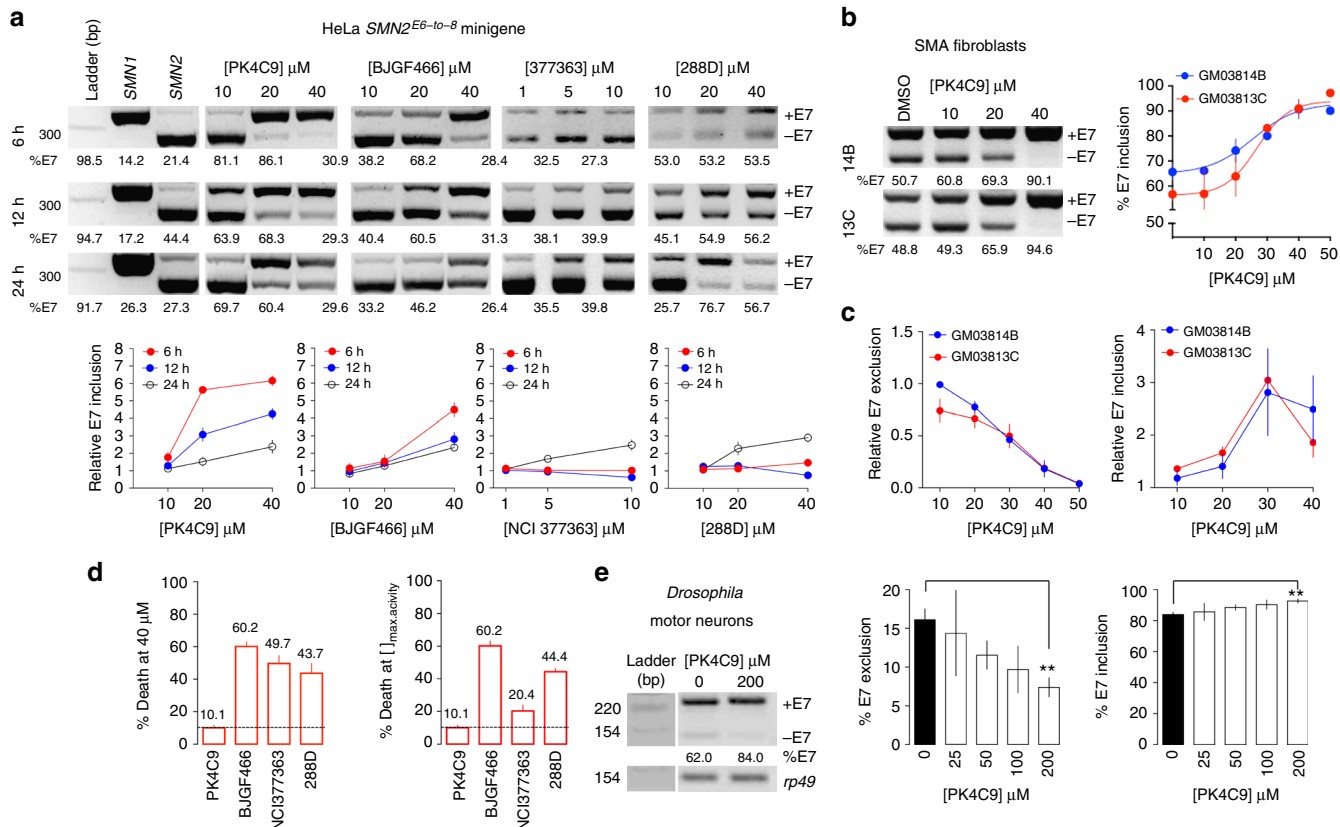

**Fig. 3** TSL2-interacting small molecules increase *SMN2* E7 splicing. **a** RT-PCR of exons 6-to-8 from HeLa cells transfected with *SMN*[E6–to–8] minigenes[16] and treated with NCI377363 (1, 5 and 10 μM), PK4C9, BJGF466 and 288D (10, 20 and 40 μM) or DMSO (0.04%, controls) for 6, 12 and 24 h (n = 6; three biological and two technical replicates). PK4C9 and BJGF466 elicit maximal E7 inclusion at early time points, which may be due to progressive compound toxicity and/or molecule secretion, as commonly seen in cancer cells. **b**, **c** RT-PCR (**b**) and qRT-PCR (**c**) showing the dose–response of PK4C9 in GM03813C (SMA) and GM03814B (parental carrier) fibroblasts after 24 h of treatment (n = 6; three biological and two technical replicates). For simplicity, the ladder is shown in **a**. In RT-PCR, *SMN2* isoforms including and excluding E7 are detected simultaneously. In qRT-PCR, *SMN2* isoforms including and excluding E7 are amplified in two separate reactions. The dose–response curve of PK4C9 reveals a narrow concentration window but achieves maximal response. Concentrations higher than 50 μM could not be measured due to poor solubility of the compound. **d** Twenty-four-hour cytotoxicity in GM03814B fibroblasts, using the lactate dehydrogenase (LDH) assay (n = 12; six biological replicates and two technical replicates). PK4C9 showed the lowest toxicity when comparing all molecules at the same concentration (40 μM; left) and the concentration of maximum splicing activity (right; PK4C9 40 μM, BJGF466 40 μM, NCI377363 10 μM and 288D 80 μM). Complete 24-h and 72-h cytotoxicity curves are shown in Supplementary Fig. 8. **e** RT-PCR from adult *D42-Gal4>UAS-SMN2*[E6–to–8] *Drosophila* flies expressing human *SMN2*[E6–to–8] in motor neurons. Flies were free fed with PK4C9 (25, 50, 100 and 200 μM) or DMSO (0.5%, controls) as larvae. *rp49* is the loading control (n = 3 biological replicates of 10–12 individuals each), \*\*p < 0.01. p-values were obtained by applying non-paired, two-tailed t tests with Welch corrections. All graphs represent mean values ± SEM

structural analogues of PK4C9 with no effect on *SMN2* splicing also failed to modify the splicing of *RPS6KB1* and *LPIN1* (Supplementary Fig. 7).

Taken together, these results validate the ability of PK4C9 to increase functional SMN protein levels and reverse molecular alterations caused by SMN deficiency in SMA cells.

**PK4C9 improves accessibility of the 5′ ss of *SMN2* E7**. To further explore the mechanism of action of PK4C9, we next used our RNA-seq data to search for shared RNA motifs at the exon/intron junctions of the 41-cassette exons most strongly affected by PK4C9 (delta PSI > 0.65), assuming that part of these were direct off-targets. Motif-finding algorithms Gibbs, MEME, Weeder and Homer were applied to our RNA-seq data, and the AGGTAAG sequence was identified as the most enriched motif at the 5′ ss of PK4C9-sensitive exons (Fig. 5a). This motif closely resembles the GAGTAAG sequence of the 5′ ss of *SMN2* E7, with which TSL2 overlaps. A battery of *SMN2* minigenes carrying structural mutations in this region of TSL2[16] were transfected in

HeLa cells in order to confirm the importance of this area for PK4C9 activity. In particular, we found that strengthening local base pairing in the 5′ ss portion of TSL2 (mutations U2–C2 and U3A17–G3C17) significantly reduced the effect of PK4C9 on *SMN2* splicing. Conversely, weakening base pairing in this region (mutation U2–A2) increased it, altogether suggesting that PK4C9 requires access to the 5′ ss residues within TSL2 in order to be active (Fig. 5b, c).

To determine whether these observations reflected a direct interaction of PK4C9 with the 5′ ss portion of TSL2, we next studied the binding of PK4C9 to TSL2 by liquid-state nuclear magnetic resonance (NMR) spectroscopy. Since high-resolution structures of TSL2 have not yet been reported in the nucleotide database (NDB), we first sought to solve the NMR structure of TSL2. A synthetic high-purity, non-labelled TSL2 RNA, as well as an adenine [13]C-labelled TSL2 RNA ([13]C-A-TSL2), were used to record a large set of NMR spectra that allowed the calculation of a bundle of 40 structures representing the TSL2 ensemble (root-mean-square deviation, RMSD, of 1.8 Å; Table 1). These

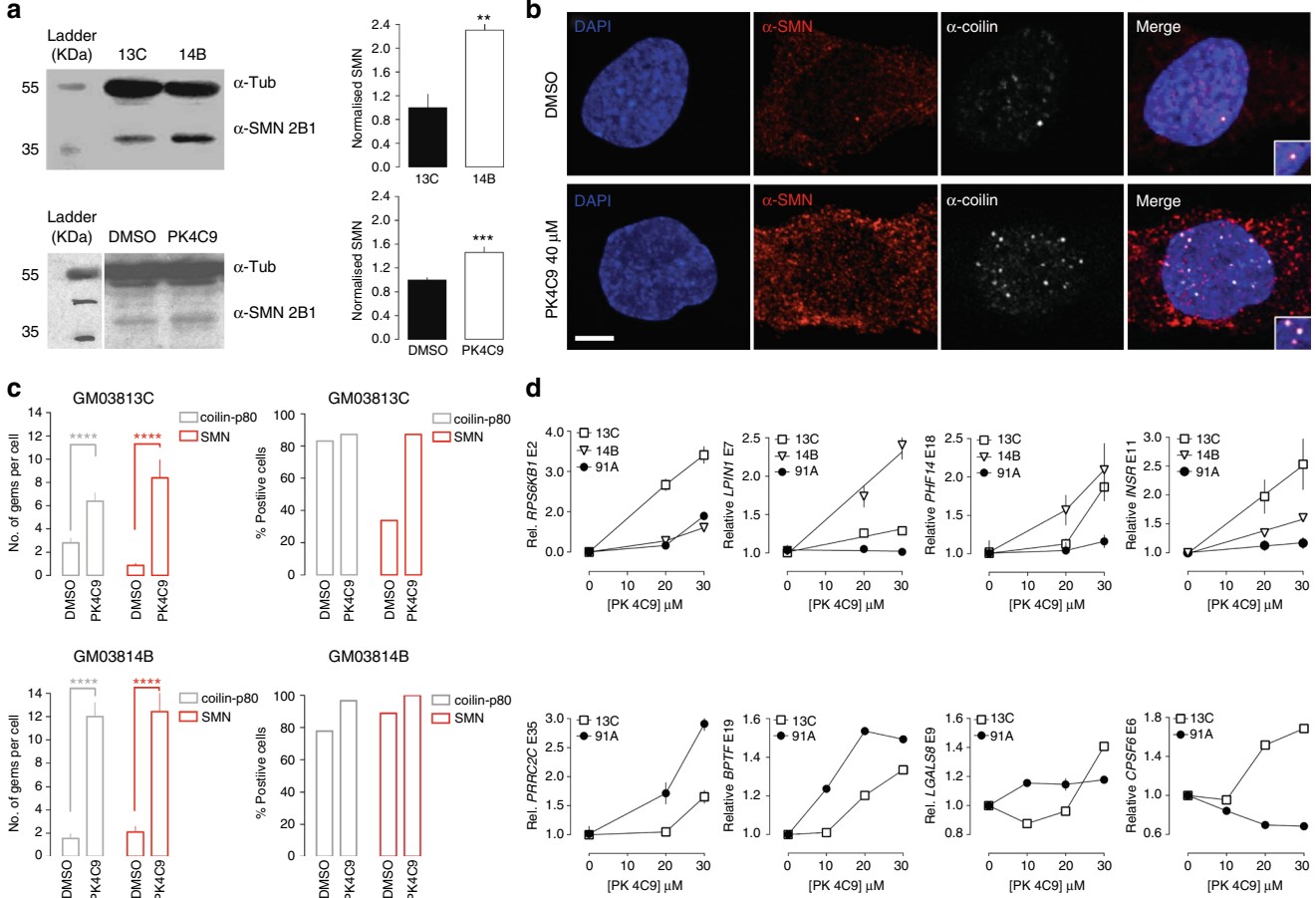

**Fig. 4** PK4C9 increases SMN protein in SMA cells. **a** Western blot quantification showing SMN levels in GM03814B vs. GM03813C fibroblasts (top) and in DMSO-treated (0.04%) vs. PK4C9-treated (40 μM) GM03813C fibroblasts (bottom) ($n = 3$ biological replicates). **b** Double-antibody staining showing the increase in SMN (red) in the cytoplasm and nucleus of GM03814B fibroblasts after treatment with PK4C9 (40 μM), and its restored localisation in nuclear gems (coilin-p80, white). Scale bar, 6 μm. **c** Quantification of the number of gems per cell (left) and the percentage of cells with gems (right) from images in the experiment (**b**) ($n = 30$ cells from three biological replicates). **d** qRT-PCR showing the effect of PK4C9 (40 μM) on the splicing of the indicated SMN-sensitive transcripts from GM03813C, GM03814B and wild-type (WT, ND36091A) fibroblasts. *RPS6KB1*, *LPIN1*, *PHF14* and *INSR* responded to PK4C9 in GM03813C and GM03814B cells but not in ND36091A, confirming that these changes are mediated by SMN recovery. *PRRC2C*, *BPTF*, *LGALS8* and *CPSF6* responded to PK4C9 in SMA and WT cells, suggesting that they are PK4C9 off-targets ($n = 8$; four biological replicates and two technical replicates). **$p < 0.01$, ***$p < 0.001$ and ****$p < 0.0001$. $p$-values were obtained by applying non-paired, two-tailed $t$ tests with Welch corrections. Graphs represent mean values ± SEM

structures showed A-helical stacking for all base pairs from A1–U19 to A8–U12, capped by a triloop consisting of residues A9–U11, with an unstable loop-closing base pair (A8–U12) that resulted in a temporary pentaloop conformation (see Fig. 6a, b; below). PK4C9 titration experiments were then performed adding up to 20 equivalents of ligand to the TSL2 RNA. Residues U2 and U19 showed the most significant PK4C9-induced chemical-shift perturbations (CSPs), clearly distinguishable from the DMSO-induced shifts (Fig. 5d). These results confirm that binding of PK4C9 affects the local conformation of TSL2 at the 5′ ss.

Given the low solubility of PK4C9 in water, the 3D structure of the TSL2-ligand complex could not be solved by NMR. Therefore, to obtain deeper atomistic understanding on the binding dynamics of PK4C9 to TSL2, we used in silico molecular docking and explicit-solvent molecular dynamics (MD) simulations in combination with our NMR TSL2 structures. Of the different binding poses of PK4C9 to TSL2 initially proposed by molecular docking (Supplementary Fig. 11), a stable binding orientation of PK4C9 to the pentaloop conformation of TSL2 was confirmed by 100-ns MD simulations (Supplementary Movie 1 and 2). In agreement with our NMR findings, this binding was mediated by

hydrogen bond interactions with residues U2 and U19, as well as π−π stacking between G18 and the indole moiety of PK4C9, and additional hydrophobic contacts (Fig. 6d, e). These interactions resulted in a partial opening of TSL2 at the 5′ ss, as quantified by the larger distance between the C1′ atoms of residues A1 and U19 (Fig. 6f) and the shorter distance between the C1′ atoms of residues A8 and U12 (Fig. 6g), as well as the increased mobility of these two residues in the presence of PK4C9 (root-mean-square fluctuation values, RMSF; Fig. 6h vs. 6c). The binding mode of the eight inactive structural analogues of PK4C9 was also studied (Supplementary Fig. 7). Analogues 001 and 004, which completely failed to bind to TSL2 in vitro and did not change *SMN2* splicing in cells, also failed to generate meaningful binding poses, providing experimental support to our atomistic binding model.

Collectively, our structural studies provide an atomistic explanation to the *SMN2* splicing modifier activity of PK4C9, whereby a local PK4C9-induced opening of TSL2 could improve accessibility of the 5′ ss of E7 to splicing factors.

**Triloop TSL2 conformations favour E7 splicing.** A second, indirect conformational consequence of PK4C9 binding to TSL2

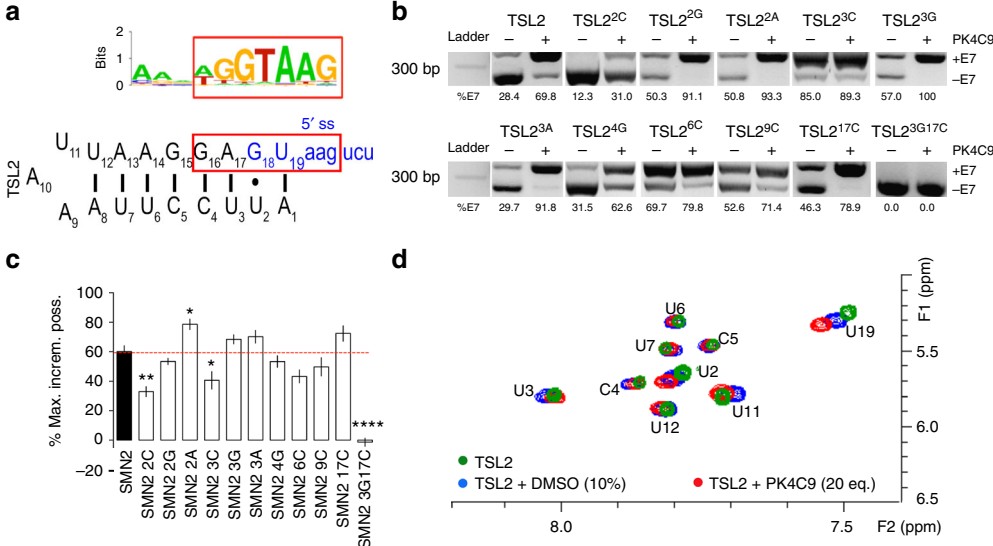

**Fig. 5** PK4C9 binding to TSL2 affects the 5′ ss of E7. **a** Motif search from RNA-sequencing data ($n = 4$ biological replicates), using algorithms Gibbs, MEME and Weeder, identified AGGTAAG as the most enriched motif in exons that are differentially spliced-in SMA cells upon treatment with PK4C9 (40 μM, 24 h) compared to SMA cells treated with DMSO (24 h). The illustration shows the example found by Weeder and the similarity with TSL2 (red box). **b** RT-PCR from HeLa cells transfected with $SMN2^{E6-to-8}$ minigenes[16] carrying n.m. TSL2 or the indicated structural mutations. Transfected HeLa cells were treated with DMSO (0.04%, controls) or PK4C9 (40 μM) ($n = 9$; three biological and three technical replicates). **c** The effect of these mutations on PK4C9 activity was measured as the percentage of maximum E7 increment possible (see Eq. 2). **d** TOCSY NMR spectrum showing the cross-peak region of H5/H6 protons of the pyrimidine bases of TSL2 (40–50 μM) with and without PK4C9 (20 equivalents). $^1$H chemical shifts were referenced to the internal DMSO signal (2.63 ppm). Residues U2 and U19 showed the most significant chemical-shift perturbations. *$p < 0.05$, **$p < 0.01$ and ****$p < 0.0001$. $p$-values were obtained by applying non-paired, two-tailed $t$ tests with Welch corrections. The graph shows mean values ± SEM

was detected in the loop part of the RNA hairpin. In particular, our MD analysis showed (1) a significant decrease in the mobility of loop residues U11 and U12 (RMSF, Fig. 6h), coupled with (2) a conformational shift from pentaloop to triloop TSL2 (Fig. 6g). These results suggested that base pairs opening at the 5′ ss end of the hairpin may inversely correlate with the mobility of the hairpin at the loop end, in order to balance the global energy of the structure, as has been described for other systems[33]. Based on this, we hypothesised that stabilizing triloop forms of TSL2 may allow more efficient E7 splicing by displaying a more accessible 5′ ss than pentaloop forms.

To test this hypothesis, structural TSL2 mutations 2A, 2C and 3G17C (Fig. 7a), which modified SMN2 E7 splicing and PK4C9 activity in transfected HeLa cells (Fig. 7b; also see Fig. 5b–c), were inserted in silico in our ligand-free NMR structure of pentaloop TSL2, and additional MD simulations were performed. Of these three mutations, 2A, which increased E7 splicing and enhanced PK4C9 activity in HeLa, showed the largest A1–U19 base pair opening (Fig. 7c, d), as well as the lowest mobility of loop residues A8–U11 (Fig. 7e; Supplementary Fig. 12). Conversely, mutations 2C and 3G17C, both of which hindered E7 inclusion and reduced PK4C9 activity in HeLa, displayed reduced A1-to-U19 distances and a proportional increase in loop residue mobility (Fig. 7c–e; Supplementary Fig. 12). Experimentally, transfecting HeLa cells with an SMN2 minigene carrying an A8U12 to 8G12C double mutation that stabilises TSL2 as a triloop, yielded an E7 inclusion of nearly 100% (Fig. 7f). Taken together, these findings establish the first link between accessibility of the 5′ ss of SMN2 E7 and TSL2 loop conformations (Fig. 7g, h).

## Discussion

RNA secondary structures are enriched at alternative splice sites[34], where they can regulate splicing by displaying splicing signals, masking them, or by placing regulatory sequences in

proximity to each other[21]. More than 150 diseases are associated to mutations affecting splicing regulatory sequences, including cancer, neurological or metabolic disorders[35–39]; some of which are known to have functional RNA structures surrounding the affected areas[40, 41]. These structures have high potential as targets to identify spicing modifier compounds, yet they remain largely underexploited. In this study, we have performed an in vitro small-molecule screening using the TSL2 RNA structure at the 5′ ss of SMN2 E7 as a biological target, providing one of the very few examples of small molecules that target the RNA structure of a splice site[42, 43]. This screening identified TSL2-binding hits that facilitate SMN2 E7 inclusion, increase SMN protein levels and revert SMA molecular phenotypes. The suitability of TSL2 as a target for the screening of small-molecule splicing modifiers was first confirmed by a combination of single-point mutagenesis and low-resolution structural experiments. These experiments demonstrated a correlation between E7 inclusion and conformational alterations in TSL2, supporting the initiative to attempt equivalent structural changes via binding of small molecules.

Recently, two series of small molecules were identified through phenotypic screenings, which reverted SMN2 splicing with high specificity[8, 9]. These studies were the first demonstration that a small molecule can be optimised to target a single splicing event, which coupled with a potentially less-challenging delivery and systemic bioavailability over other current modalities, makes small molecules a highly promising therapeutic option for SMA. Small-molecule screenings using commercial high-chemical diversity libraries are, however, generally biased for modulating protein function, yielding much lower hit rates for RNA targets[23]. To overcome this, we generated a target-focused library of small molecules with privileged scaffolds to bind RNA. Target-focused libraries provide a series of benefits, as recently reviewed[44, 45]. For example, (1) focused libraries save time and resources by reducing the number of compounds to screen, (2) yield higher hit rates by

## Table 1 NMR and refinement statistics for the TSL2 RNA

|  | TSL2 |
|---|---|
| **NMR distance and dihedral constraints** | |
| Distance restraints | |
| Total NOE | 458 |
| Intra-residue | 258 |
| Inter-residue | 200 |
| Sequential ($\lvert i - j \rvert = 1$) | 154 |
| Nonsequential ($\lvert i - j \rvert > 1$) | 46 |
| Hydrogen bonds | 14 |
| Total dihedral angle restraints | 114 |
| Sugar pucker | 50 |
| Backbone | 54 |
| Sugar to base | 10 |
| Base pair planarity | 6 |
| **Structure statistics (mean ± SD)** | |
| Violations (mean and s.d.) | |
| Distance constraints (Å) | 0.037 ± 0.002 |
| Dihedral angle constraints (°) | 0.26 ± 0.11 |
| Max. dihedral angle violation (°) | 5.57 |
| Max. distance constraint violation (Å) | 0.49 |
| Deviations from idealised geometry | |
| Bond lengths (Å) | 0.0026 ± 0.0001 |
| Bond angles (°) | 0.75 ± 0.03 |
| Impropers (°) | 0.59 ± 0.04 |
| Average pairwise r.m.s. deviation[a] (Å) | |
| All RNA heavy | 1.85 ± 0.58 |
| Stem heavy (residues 1–7, 13–19) | 1.23 ± 0.35 |
| Loop heavy (residues 8–12) | 1.54 ± 0.57 |

[a]Statistics from a final bundle of 40 structures after water refinement

eliminating compounds that are unlikely to bind to the target[23, 46] and (3) can reduce the hit-to-lead timescale, given that the properties of their compounds have already been filtered to suit the type of target in question. From the screening of our focused library, 19 TSL2-interacting hits were further examined for their ability to modify *SMN2* E7 splicing. Nearly half of them successfully promoted exon inclusion, thus validating our target choice and screening strategy. Particularly, marine natural molecule homocarbonyltopsentin (PK4C9) showed the most promising effect, reverting molecular phenotypes in HeLa cells, type-I SMA-derived fibroblasts and *Drosophila* motor neurons, demonstrating the translatability of our results to different cell types and models. In SMA cells, PK4C9 increased E7 inclusion by ~40%, coupled with a 1.5-fold increase in SMN protein. Similar increases in SMN levels have been shown to be sufficient to reverse SMA phenotypes in mice models (including lifespan and motor function[8, 9]) and allow transitioning from severe to mild forms of SMA[4], providing proof-of-principle to the use of TSL2-modifying small molecules in SMA drug discovery, as well as encouraging the chemical optimisation of PK4C9.

A number of structural analogues of PK4C9 were designed and synthesised in this study, which displayed different TSL2-binding efficiencies. These efficiencies significantly correlated with their effect on E7 splicing. Moreover, structural mutations in TSL2 also affected the activity of PK4C9, altogether demonstrating that the cellular activity of PK4C9 is mediated by its interaction with TSL2 (see Supplementary Table 1). The TSL2 mutations that most strongly modified PK4C9 activity involved the GAGTAAGT motif of the 5′ ss of E7, which was identified by our RNA-seq and structural data as the target site of PK4C9. In a recent report, the GAGTAAGT motif of *SMN2* E7 was also involved in NVS-SM1 and NVS-SM2 activity, two *SMN2* splicing modifiers identified through phenotypic screening[9]. However, these molecules carried out their effect on E7 splicing via directly affecting the binding of

the splicing machinery to the 5′ ss. The fact that mutations located in the 5′ half of TSL2 (f.e., 2A or 2C) can modify PK4C9 activity rules out a direct interaction with 5′ ss-binding factors, and confirms a conformational change of TSL2 as responsible for the effect of this molecule on E7 splicing.

A series of studies in mice have shown that reduction of SMN protein results in widespread splicing abnormalities[29–31, 47]. Recovery of SMN is therefore expected to lead to a large number of splicing changes, which would represent the reversal of at least part of such splicing abnormalities. Our RNA-seq analysis detected 201 differentially spliced genes in SMA cells upon treatment with PK4C9, some of which were linked to SMN rescue. Only two other examples of *SMN2* splicing modifying small molecules exist in the literature for which RNA-seq data are also available[8, 9]. Three differentially spliced genes (*SMN2*, *SLC25A17* and *VPS29*) are common between these two molecules and PK4C9, further supporting that at least some of all PK4C9-induced splicing changes are the consequence of SMN rescue. PK4C9-sensitive splicing changes also included off-targets, which could account for the 10% toxicity of PK4C9 in GM03814B fibroblasts. Being able to discern between undesired off-target vs. SMN recovery-mediated splicing changes is key for the chemical optimisation of PK4C9. Such optimisation would in turn be aided by our structural results. Our NMR and MD studies determined that PK4C9 binding to TSL2 requires, and promotes, a partial opening of TSL2 residues U2 and U19, which resemble the conformational effect of mutation 2A. In the case of NMR, the PK4C9-induced CSPs were of small magnitude, due to the fact that G18 and U2 imino protons are poorly visible to NMR (fast water exchange). However, our MD clearly confirmed stable interactions between PK4C9 and TSL2 residues U19, U2 and G18. Residues U2 and G18 are part of the non-canonical G·U wobble base pair of TSL2, which are known to offer unique structural and ligand-binding properties[48], and pose the ideal starting point for lead optimisation.

TSL2 was first described as a triloop by in vitro enzymatic probing, whereas a latter study using in vitro SHAPE found it in the pentaloop form[16, 17]. Here, we show that both species coexist and use NMR and molecular dynamics (MD) to provide an atomistic explanation as to how the TSL2 equilibrium between pentaloop and triloop conformations influences E7 splice site recognition[49]. In particular, we could associate the triloop form of TSL2 to a more efficient E7 splicing. Taking advantage of this finding, bioactive small molecules could now be rationally designed that target not only TSL2 at the 5′ ss level, but also at the hairpin loop, stabilising it as a triloop (examples of RNA-loop targeting molecules can be found here[18, 50]).

In summary, our study contributes to the increasing use of small molecules to rationally target RNA[18–20, 51–56], and opens new avenues for rational drug discovery in SMA, setting a precedent for other splicing-mediated disorders, where the relevant RNA structures could be similarly targeted to modify the outcome of the splicing events that they regulate.

## Methods

**Fluorescence displacement screening**. RNA (0.5 μM) was snap annealed in 8 mM $Na_2HPO_4$ at pH 7, 185 mM NaCl, 0.1 mM EDTA and 40 μg/mL BSA, and incubated per quadruplicate with TO-PRO-1 (1 μM), and DMSO (control) or ligand (100 μM, screening) for 30 min in black 96-well plates. The plate was excited at 485 nm and fluorescence emission was collected at 528 nm in a Synergy$^{TM}$ Mx (BioTek) plate reader. Molecules that decreased TO-PRO-1 fluorescence intensity by >20% were considered positives, as per equation (1)

$$\text{Activity (\%)} = 100 - \left[ \left( \frac{A - B}{C - D} \right) \times 100 \right], \qquad (1)$$

where (*A*) indicates fluorescence intensity of TO-PRO-1 with RNA and compound, (*B*) fluorescence intensity of TO-PRO-1 with compound, (*C*) fluorescence intensity

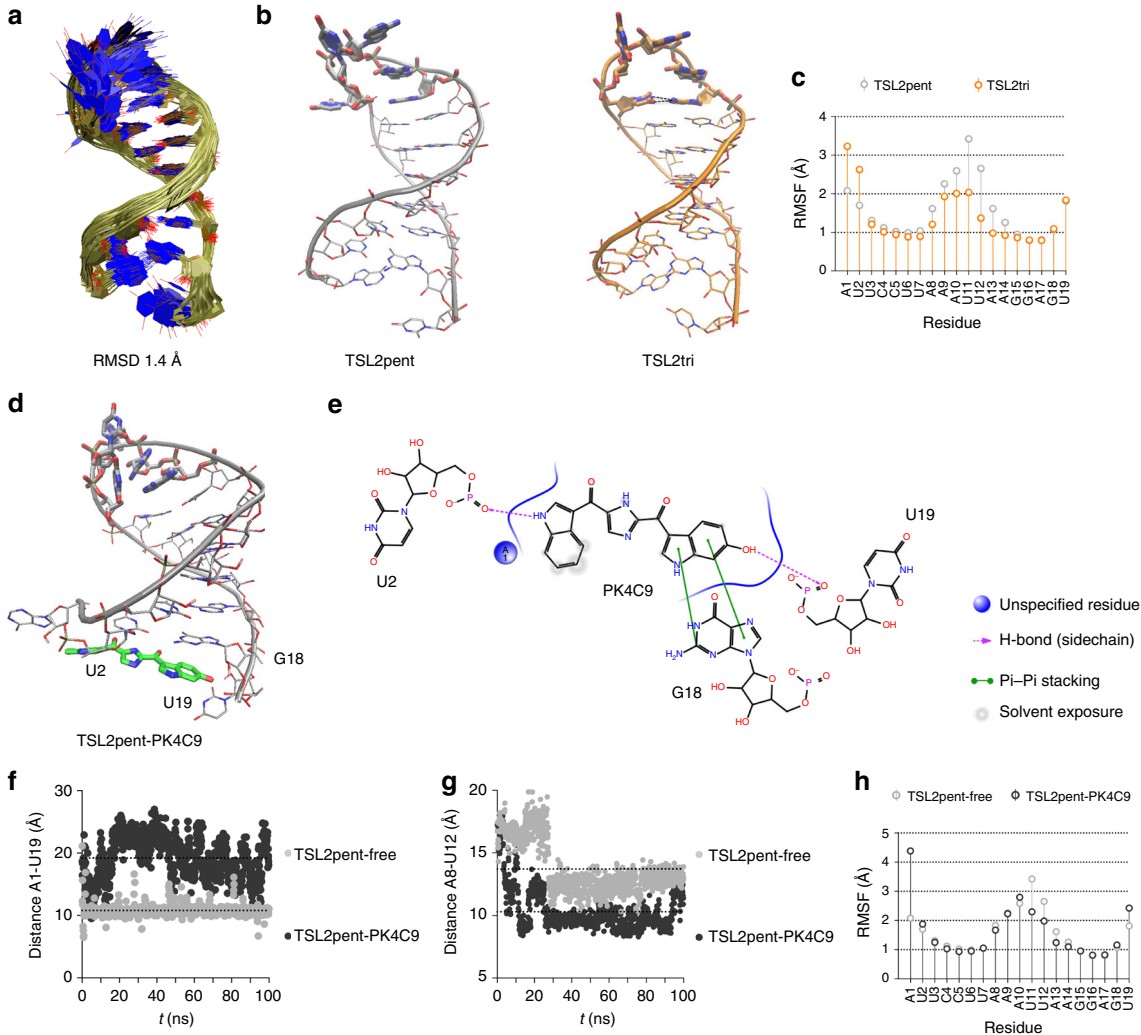

**Fig. 6** PK4C9 directly causes opening of 5′ ss TSL2 residues and indirectly induces triloop conformation. **a** Bundle of 40 NMR structures of TSL2. **b** Representative ligand-free pentaloop (TSL2pent, silver) and triloop (TSL2tri, orange) NMR structures of TSL2. **c** Comparison of the mobility of TSL2pent and TSL2tri residues in 100-ns molecular dynamics (MD) trajectories, measured by root-mean-square fluctuation (RMSF). **d** Representative model structure of the binding mode of PK4C9 (green) to TSL2pent (TSL2pent-PK4C9, grey), taken from the most abundant cluster of a 100-ns MD trajectory. **e** Close-up 2D view of the binding mode of PK4C9. Interactions were plotted using the Maestro software. **f** Graph showing the PK4C9-induced increase in the distance between the C1′ atoms of residues A1 and U19 throughout the MD trajectory, as an indicator of terminal TSL2 opening. **g** Graph showing the PK4C9-induced decrease in the centre of mass distance between residues A8 and U12 throughout the MD trajectory, as an indicator of triloop formation. **h** Per-residue comparison of the mobility of TSL2pent-free and TSL2pent-PK4C9, measured by root-mean-square fluctuation (RMSF)

of TO-PRO-1 with RNA and (D) intensity of the TO-PRO-1 alone. When Ribo-Green was used instead of TO-PRO-1, 300 nM of dye was used, with fluorescence emission read in a LightCycler® qPCR instrument using Sybr Green filters.

**Cell culture**. HeLa cells were a gift from Prof. U. Rüegg (University of Geneva, Switzerland). HeLa cells were cultured in DMEM medium with 10% FBS and 1% antibiotics (penicillin and streptomycin). ND36091A, GM03814B and GM03813C fibroblasts were obtained from the Coriell Institute for Medical Research, and grown in MEM medium with 15% FBS, 1% antibiotics (penicillin and strepto-mycin) and 2 mM glutamine (freshly added). All lines were kept in a humidified incubator at 37 °C in 5% CO$_2$. For all RNA and protein extractions, $4 \times 10^5$ cells were seeded per well per triplicate in six-well plates (volume 2 mL) and grown to 80% confluence. DNA transfection of pCI-SMN2$^{E6-to-8}$ minigenes[16] in HeLa was performed using X-tremeGENE HP Transfection Reagent (Roche Life Science) and 1 µg of plasmid DNA. After 24 h, treatment was added to the wells. In non-transfected cells (fibroblasts), compounds were directly added to the wells once 80% cell confluence was reached. Fibroblasts and HeLa were regularly checked for mycoplasma contamination.

**Drosophila**. A total of 10–12 first-instar larvae expressing the UAS-SMN2:luc minigene under the control of the D42 promoter (D42-Gal4, Bloomington #8816) were transferred per triplicate into tubes containing 0.5% DMSO (control) or

PK4C9 in 0.5 mL of standard nutritive media. Larvae were free fed with PK4C9 for 5 days. Tubes were kept at 25 °C and flies were collected 24 h after hatching for homogenisation.

**Reverse transcription (RT)-PCR**. Total RNA was extracted using the RNeasy mini kit with on-column DNase digestion (Qiagen). Between 0.5 and 1 µg of RNA were used for reverse transcription with Super Script II (Invitrogen) or the High-Capacity cDNA Reverse Transcription Kit (Applied Biosystems), using random hexamers (cultured cells) or a gene-specific primer 5′-CAGCGTAAGT-GATGTCCACCT-3′ (Drosophila). A total of 100 ng of cDNA were used as tem-plate in semi-quantitative PCR with GoTaq polymerase (cultured cells; Promega) or 2×PCR Super Master Mix (Drosophila, Biotool). Three biological replicates and three technical replicates per biological replicate were obtained, unless stated otherwise. Primer sequences, and PCR conditions are described in Supplementary Table 4. Bands were resolved in 3% agarose gels at 4 °C for 90 min. Representative full agarose gel images supporting our main findings are shown in Supplementary Fig. 13. Quantification of band intensity was performed on ImageJ. The effect of TSL2 mutations on PK4C9 splicing modifier activity is shown as the percentage of maximum E7 increment possible (% MIP), defined by Eq. (2):

$$\% \text{ MIP} = \left[ \left( \frac{\text{E7}_{\text{PK4C9}} - \text{E7}_{\text{DMSO}}}{100 - \text{E7}_{\text{DMSO}}} \right) \times 100 \right]. \tag{2}$$

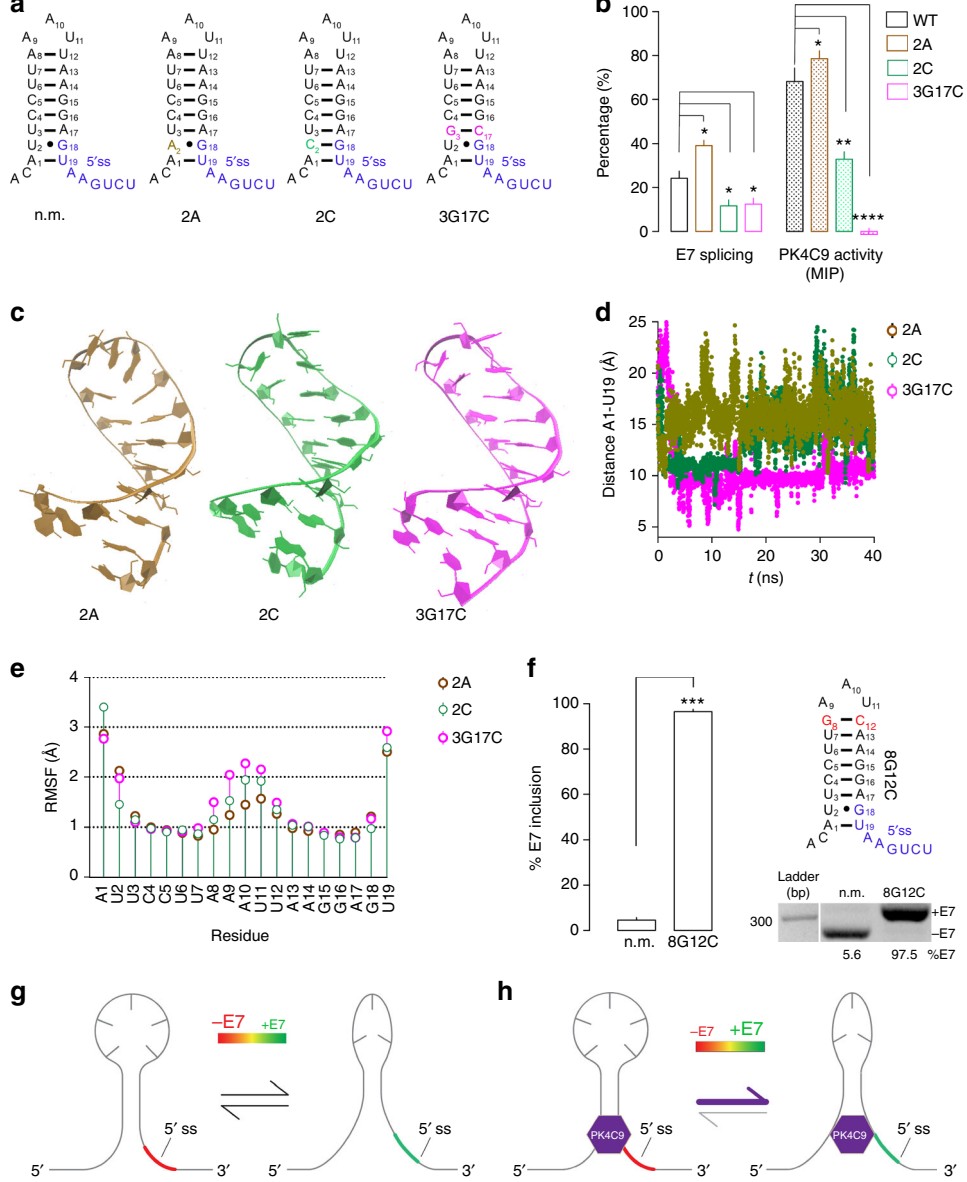

**Fig. 7** The triloop conformation of TSL2 facilitates *SMN2* E7 splicing. **a** Sequence of non-mutated (n.m.) TSL2 or TSL2 carrying the 2A, 2C, or 3G17G mutations[16]. **b** Graph recapitulating that mutation 2 A increases E7 splicing and PK4C9 activity (expressed in percentage, %) in HeLa cells transfected with *SMN2*$^{E6-to-8}$ minigenes; whereas 2C, and 3G17C decrease them (see also Fig. 5b, c). MIP maximum E7 increment possible (see Eq. 2). **c** Representative structures of the 2A, 2C and 3G17C RNAs taken from their 40-ns MD trajectories. 2A TSL2 exists mainly as a triloop, whereas 2C and 3G17C TSL2 exist mainly as pentaloops. **d** Quantification of the distance between the C1′ atoms of residues A1 and U19 throughout the MD trajectories, as an indicator of terminal TSL2 opening. The 2A structure shows the largest opening, whereas the 3G17G structure shows the shortest one. **e** Per-residue comparison of the mobility of the 2A, 2C and 3G17C mutant structures (RMSF). The mobility of loop residues A8–U11 is the highest in the 3G17C structure and lowest in the 2A mutant. **f** RT-PCR from HeLa cells transfected with an *SMN2*$^{E6-to-8}$ minigene carrying either n.m. TSL2 or the 8G12C double mutation. The 8G12C mutation, which stabilises TSL2 in its triloop conformation, increased E7 splicing to nearly 100% (*n* = 3 biological replicates). **g, h** Schematic model of the proposed mechanism of action of PK4C9. Triloop and pentaloop conformations of TSL2 coexist, with triloop conformations displaying better access of the 5′ ss and E7 splicing. PK4C9 would bind to pentaloop TSL2 directly improving 5′ ss exposure and indirectly causing a shift to the triloop (**h**). *$p < 0.05$, **$p < 0.01$, ***$p < 0.001$, ****$p < 0.0001$. *p*-values were obtained by applying non-paired, two-tailed *t* tests with Welch corrections. The graph shows mean values ± SEM

**qPCR.** A total of 25 ng of cDNA were used as template for qPCR with the TaqMan Fast Advanced Master Mix (Applied Biosystems), using the default fast mode of a StepOnePlus Real-Time PCR System (Applied Biosystems). *GAPDH* was used as the endogenous control (5 ng of cDNA template) after validating its stability compared to other housekeeping genes. Three biological replicates and three technical replicates per biological replicate were obtained, unless stated otherwise. Primer and Taqman probe sequences and concentrations are described in Supplementary Table 4. The relative expression of PK4C9-treated samples to *GAPDH* and to the control group (DMSO) was obtained by the $2^{-\Delta\Delta Ct}$ method. Due to the low copy number of *SMN2* transcripts, non-radioactive northern blot could not be

used as a validation technique. *SMN2* isoform bands are shown by semi-quantitative PCR instead.

**Western blot.** A total of 25 μg of total protein from whole-cell extracts from three biological replicates (40 μM PK4C9 or 0.04% DMSO, 48 h) were fractionated in 12% SDS-PAGE and transferred to a PVFD membrane for 1 h in a cooled tank. After blocking with 5% BSA, the membrane was washed (TBST) and incubated with primary antibodies anti-α-Tub (1:2000, endogenous control; Sigma, cat # T9026) and anti-SMN (1:500; clone 2B1; Sigma, cat # S2944) for 2 h at RT.

Membranes were washed and incubated with a secondary DyLight 680-conjugated anti-mouse antibody (1:10,000; Cell Signaling Technology, cat # 5470) for 2 h at RT. Membranes were washed and signal visualised with a LI-COR Odyssey Imaging System in the 700-nm channel. Representative full-blot scans supporting our main findings are shown in Supplementary Fig. 13.

**Immunohistochemistry.** A total of $4 \times 10^5$ cells were seeded in 2 mL per well in two-well chamber slide systems (Lab-Tek), and kept at 37 °C in 5% $CO_2$. At 80% confluence, each well was treated for 24 h with DMSO (control) or PK4C9 (40 μM). Cells were washed (1×PBS) and fixed with 4% PFA for 10 min at RT. Permeabilisation was carried out with 0.25% Triton X-100 in PBS at RT for 10 min, followed by washes. Cells were blocked with 1% BSA in PBST (0.1% Tween 20) for 1 h, and incubated with anti-SMN (1:500; clone 2B1; Sigma, cat # S2944) and anti-coilin-p80 (1:50; H-300 Santa Cruz Biotechnology Inc., cat # sc-32860) o/n at 4 °C. After washing, Alexa Fluor 647-conjugate anti-rabbit (Molecular Probes, cat # A-31573) and Alexa Fluor 555-conjugate anti-mouse (Molecular Probes, cat # A-21422) secondary antibodies were used at 1:500 for 90 min at RT. Slides were rinsed and mounted on Vectashield mounting medium with DAPI (Reactolab). All images were taken under the same settings in a confocal Zeiss LSM700 microscope. A total of 30 cells from three biological replicates were analysed.

**Statistics.** All statistical analyses were performed using GraphPad Prism unless otherwise stated. Two-tailed, non-paired $t$ test was used for comparisons between two groups, with Welch correction in the case of unequal variances. For the small-molecule screening, a cut-off value of 20% FD was applied; followed by two-tailed, non-paired $t$ tests with Bonferroni correction for multiple comparisons. Data are presented as means ± SEM in all cases except Table 1. Significance values are detailed in the figure legends.

**Code availability.** No computer codes were developed for this study. Input scripts and software alterations used for NMR structure calculation, molecular dynamics or RNA-seq analysis are described in Supplementary Methods and are available from the corresponding authors upon request.

**Data availability.** The NMR structure of TSL2 was deposited in the PDB (accession number 5N5C) and the BMRB (accession number 34100). Our RNA-seq data have been deposited in the Gene Expression Omnibus repository (accession code GSE94111). Additional data that support the findings of this study are available from the corresponding authors upon reasonable request.

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

## Acknowledgements

We thank members of the LS group Dr. S. Tardy and O. Patthey; member of the RA group E. Cerro and Creoptix (www.creoptix.com) for technical assistance; Prof. P. Moreau (University of Clermont Auvergne, FR) for providing compounds; Prof. R. Singh (Iowa State University, US) for providing *SMN2* minigenes; Prof M. Zhang (Tufts University, US) for providing the *SMN2* minigene used in *Drosophila* and Dr. E. C. O'Connor (University of Geneva, CH) for input on the manuscript. This work was supported by the University of Geneva and grants from the Swiss National Science Foundation (SNSF) (Sinergia, CRSI33-130016) to L.S; Protein Kinase Research (Pro-Kinase, LSHB-CT-2004-503467) to L.S. and P.G.; the European Molecular Biology Organization (EMBO, ALTF 253-2012) and the Schmidheiny Foundation to A.G.-L.; SMA Europe to A.G.-L. (17623) and R.A. (19243); BioNMR and iNEXT to A.G.-L. and H.S.; the German Research Foundation (DFG, CRC902) to H.S. and A.W.; the state of Hesse through institutional funds for BMRZ to H.S. and C.R.; LOEWE programme SynChemBio to H.S. and H.R.A.J.; the ERC Council (SimDNA) and the Spanish Ministry of Science and Competiveness (BFU2014-61670-EXP, BFU2014-52864-R) to M.O. and the Generalitat Valenciana (Santiago Grisolía PhD programme) to P.K.

## Author contributions

A.G.-L. and L.S. conceived the project. A.G.-L. performed and analysed the screening, low-resolution structural experiments, and all cell culture assays, with help from D.S., O. P. and R.P.; F.T. performed and analysed the computational modelling with help from I. F., A.G.-L. and L.S.; G.C. designed the in silico database that was filtered to generate the RNA-binding small-molecule library; H.R.A.J., A.W. and C.R. performed and analysed the NMR experiments with help from A.G.-L.; A.C., G.F., P.G. and B.J. synthesised compounds; N.B., R.S., K.H. and M.E. performed the biostatistical analysis of RNA-seq data, and P.K. performed and analysed the experiment in *Drosophila*. F.M., M.O., R.A., B. J., H.S. and L.S. supervised their contributions to the study. A.G.-L. wrote the paper with feedback from all authors.
