## [Peer Review File · Nature Communications]

Reviewers' Comments:

Reviewer #1:

Remarks to the Author:

The manuscript of Garcia Lopez et al describes the identification of a small molecule (PK4C9) as a modulator of SMN exon 7 splicing through its interaction with the TSL2 RNA loop located on the 5'ss of exon 7. The data is well described and the authors present a thorough study of the mechanism of action of PK4C9.

Specific comments are as follows:

1- In Fig 1b the visual estimation of the intensity of the western bands not always match with the scanning data presented in Fig 1c. In particular for the mutant TSL2 4G (5th lane). In fact, while in the western I would guess is even less E7 inclusion that in the SMN 2 TSL2nm (second lane) in the scanning appears slightly higher. An inconsistency between western and scan may also be present with the mutant 9C.

In Fig 1e the meaning of "others" should be explained in the figure legend

2- Page 5 lines 112-114 The authors interpret the effect of the 4G mutation in structural terms, they should discuss also the possibility of creating a splicing factor binding site that in turn inhibits splicing of exon 7 independently of the structure. The 4G mutation seems to expose a sequence GUAAGGAGU that although not fully "canonical" could be a binding site of hnRNPA1 or other proteins of that family whose binding will in turn hamper 5'ss recognition

3- Minor point, there is a typo in Page 7 line 148 TSL2-interacting

4- Fig 3e and Page 7 lines 170-176: The authors should comment on the fact that in Drosophila the SMN2 gene exon 7 seems to be included in a very high percentage (72%?). In fact exclusion of exon7 is barely visible before PK4C9 treatment

5- Fig 4a and Page8 lines 182-187: It is surprising that with the almost total restoration of normal exon 7 inclusion at the RNA level (Fig 3b and 3c) there is such a marginal increase in protein as seen by western blot. The recovery of the gems (Fig 4b) is certainly impressive but a measure of the total amount of mRNA containing exon 7 before and after treatment with PK4C9 is worth doing. The qPCR tends to give overestimates of the functional intact mRNA present in the cell, a Northern blot will give a more definitive result of the mRNAs including and excluding exon 7.

6- It is not clear how the authors differentiate PK4C9 effects on splicing due to SMN recovery or to a direct off target effect of PK4C9. The eventual off target effects seen in WT fibroblasts after PK4C9 treatment should be more widely discussed particularly on the light of potential use as a therapeutic agent.

Reviewer #2:

Remarks to the Author:

In this manuscript Garcia-Lopez et al. identified a natural compound binding to the stem-loop RNA structure of SMN2 exon 7 and report a detailed characterization of the mode of action of this compound in vitro, in cells, and in drosophila. The manuscript contains a large amount of data, is technically sound, and a piece of work of high quality.

However, there are several points that need to be addressed before the manuscript is suitable for publication. This regards in particular the fact that the compound binds to the terminal ends of the stem loop structure. In the in vivo situation the terminal ends are not freely available. Is the interaction mode determined by NMR and MD compatible with this in vivo situation? From figure 6 d and 6 there seem to be only a few contacts between the natural compound and the RNA.

Wouldn't the authors expect a broad range of unspecific effects?

The NMR chemical shift perturbations shown in Figure 5d seem rather small to me given that the compound has aromatic rings. Wouldn't the authors expect larger chemical shift perturbations due to ring current effects. Do the chemical shifts back-calculated from the MD model match what they see in the spectrum? Did the authors follow the chemical shift changes of the ligand by NMR and could they validate their MD data with respect to the ligand - stem-loop RNA interactions shown in Figure 5?

Minor points:

- some figure labels are too small and hard to read, e.g. Fig. 1a numbering of nucleotides etc.
- some figures miss proper axes labels/units, e.g. Figure 1d (what is 'max'), Figure 1f (CD)
- in some figures a broken axis is shown, but there are no data points extending to these values (e.g. Fig. 1e, Fig. 1d)
- some figures miss error bars, e.g. Fig. 1d,
- Figure 3e 'Drosophila MN' is there anything missing?
- Figure 4 seems of poor resolution

Reviewer #3:

Remarks to the Author:

Garcia-Lopez et al show that one can use drugs (PK4C9 as most efficient) to rescue the reduction of SMN1 by mis-splicing, causing SMA, by induction of a secondary structure change of the isoform SMN2 becoming splicing competent at the required point. In this paper many methods are coherently represented and a comprehensive evidence based conclusion is drawn.

Most methods are standard in the field, as is most of the application. However, the detailed structural work for finding out the mechanism of drug action is unusual and brings out new information about this drug and the splicing mechanism - at least partial - some of this was already suggested at lower resolution. This work is well done and should set a standard for the field of drug development on RNA drugs - to include more detailed mechanistic studies with the completeness of methods.

Major points should be clarified:

- Line 275-277: Why is this coupling assumed? (loop formation to terminal opening) -> how is this interaction/conformational change information transported over that long distance?
- Could some of it simply be explained by less overall stability (e.g. U2A mutation) -> add melting temperatures (e.g. from Fig S2, but preferred by UV),
- Line 129-132: would that not also decrease specificity?

Some minor changes need to be addressed:

- line 114/115: what indicates the native page conclusion of hairpin formation? – more detail please (e.g. marker...)
- CD experiments: melting curves: maybe the effect is caused by difference in stability rather than conformation?
- Suppl. Table S1 is wrongly called Suppl. Table S2, Tables S3,4 & 6 are missing or wrongly assigned -> Supplementary tables are a mess
- Line 163 & 166: please define what control cells represent (cells with SMN1 basal mRNA levels? Or just the same cells – DMSO influence – as mentioned in line 194?) -> for immunohistochemistry it is mentioned what control is as cells – not for drug testing (line 433) and for Figure 5 (line 640)
- Fig 5d: CSP are significantly different between DMSO and PK4C9, however, they have the same

- direction, indicating the same underlying structural change (potentially in this case destabilization)? – please explain -> U11 different trend
- Fig 6g: U19 data missing, hence line 273 not possible to observe (U19 also missing in 6C)
 - Please make an overview table over mutants and effects (E7 inclusion, terminal opening, tri/penta-loop distribution, PK4C9 activity, stability)
 - Line 148 TSL-2 interacting
 - Line 376 "one of the few examples ..."
 - Suppl. line 280, comparisson (with just one s)

Reviewer #4:

Remarks to the Author:

Garcia-Lopez et al describe the identification of a novel small molecule binder of the TSL2, RNA stem loop structure in the SMN2 gene. Using a combination of NMR, Next generation sequencing and mutagenesis studies they go on to show that PK4C9 binds to TSL2, promoting a conformational shift that favors increased inclusion of SMN2 exon 7.

The current study represents the first study describing the identification of a small molecule modulator of TSL2 but the importance of TSL2 as a key element in regulating SMN2 splicing has been previously well documented (Singh et al; NAR, 2007) and is recognized as an attractive target in the field. Additionally, the idea of using small molecules to target RNA secondary structure has attracted a lot of attention and has clear precedent (e.g Velagapudi et al; NCB 2013; Childs-Disney et al; ACS Chem Biol 2014; Velagapudi et al; PNAS, 2016; Patwardhan et al, MedChemComm 2017). Hence, the current study does not offer any significant advancement in our understanding of achieving SMN2 splicing modulation.

1. While the authors performed a screen for TSL2 interactors and identified their top hit PK4C9 using this approach, they fail to provide any compelling evidence for the selectivity of their hits. It is critical that they counter-screen their hits on unrelated, control RNA secondary structures to provide evidence that PK4C9 and related hits do not act as promiscuous RNA binders.

2. The binding activity is only demonstrated indirectly using fluorescent techniques, well-known to be prone to false positives, without any counter-screen data. The structural aspect of compound binding is simulated based on RNA NMR data, virtual docking and molecular dynamics. While the authors do show NMR shifts of RNA when compounds are added, these could well be conformational changes in RNA which could be induced by metal ion contaminations in the compound or by metal ion chelation by the compound (the compound does resemble a chelator). Can they rule out this possibility? Can they show shifts in compound spectra as well? Bottom line is that they show no evidence for direct, selective binding such as SPR and ITC. It is imperative that they provide direct evidence of compound binding to TSL2 using independent, label-free biophysical methods as suggested above. They should also provide an assessment of binding selectivity using control RNA structures when carrying out these assessments.

3. Do the authors have any evidence for dose responsiveness of their compound in the SMN2 mini-gene or SMN protein assays? The effect in the minigene assay and in the SMA patient fibroblast splicing assessment (Fig 3) looks like an all or none response. The protein increase in Fig 4a is very modest at the single dose (40uM) for which data is shown. In the minigene assay (Fig 3 a) it looks like PK4C9 and BJGF466 elicit maximal exon 7 inclusion at early timepoints and the effect tends to get weaker at the 24 hr time-point. Do the authors have an explanation for this? Have they looked at later time points?

4. A time course (up to 72 hrs or longer) study in dose response format is needed to make a confident statement about the cytotoxicity of these molecules. A 24 hr cytotoxicity study as shown

in Fig 3d is inadequate and a tad misleading although it shows PK4C9 to be superior to the other tested molecules at one early timepoint and at a given dose. Given that there are over 200 splicing events impacted by the molecule a more thorough evaluation of cytotoxicity is warranted.

5. In their RNA-Seq experiment the authors identified 290 transcripts with modified splicing relative to DMSO. The scope of alternative splicing events impacted by the compound may be an underestimate given that sequencing reads could not unambiguously map to full length transcripts. A more stringent statistical assessment of the splicing changes and a rank ordering of the changes based on significance would be very informative. Also, It would be good for the authors to clarify which of the splicing events are due to rescue of SMN2 splicing and which may be resulting from non-specific interactions of the compound with other RNA sequences or RNA secondary structures, genome wide. Comparison of the RNA Seq profile of PK4C9 in wild type versus SMA patient fibroblasts could offer insights on this front. Alternatively, comparison to SMN overexpression / rescue or to Spinraza treatment would be informative in this regard.

6. The authors should seriously consider including a structurally related, inactive (in SMN assays) compound as a negative control in their key cellular and biophysical studies, NGS etc

7. In recent years there has been significant progress in identifying small molecule and antisense-oligonucleotide based approaches to enhancing SMN2 splicing / exon 7 inclusion. Almost all of these approaches have relied on a couple of mouse models of SMA to demonstrate in vivo efficacy. The current study however provides evidence of in vivo efficacy in a less commonly used fly model of SMA. This does not allow for proper benchmarking / comparison of TSL2 modulators to previously demonstrated approaches to enhancing SMN2 exon7 splicing that are currently in the clinic, which is critical given the current state of the field.

While Garcia-Lopez et al present promising, early evidence for the identification of small molecule modulators of the TSL2 stem loop structure in SMN2 the study fails to provide thorough validation and selectivity assessment of the compound(s). The current study represents a modest, incremental increase in our structural and mechanistic understanding of how TSL2 (which has long been known to be a key regulatory region for SMN2 splicing) may be modulated with small molecules to enhance SMN2 exon 7 inclusion.

In summary, I would not recommend accepting this manuscript for publication in its current form.

**Reviewer #1**

We thank the reviewer for their time and constructive comments. We have performed
additional experiments and text modifications to address their concerns, as detailed
below. To aid reading of the revised manuscript, all major changes have been
highlighted in yellow.

**Remarks to the Author:**

The manuscript of Garcia Lopez et al describes the identification of a small molecule
(PK4C9) as a modulator of SMN exon 7 splicing through its interaction with the TSL2
RNA loop located on the 5'ss of exon 7. The data is well described and the authors
present a thorough study of the mechanism of action of PK4C9.

Specific comments are as follows:

**1- In Fig 1b the visual estimation of the intensity of the western bands not always**
**match with the scanning data presented in Fig 1c. In particular for the mutant**
**TSL2 4G (5th lane). In fact, while in the western I would guess is even less E7**
**inclusion that in the SMN 2 TSL2nm (second lane) in the scanning appears slightly**
**higher. An inconsistency between western and scan may also be present with the**
**mutant 9C.**

First, we would like to clarify that that Fig. 1b represents RT-PCR products resolved by
agarose gel electrophoresis, not Western Blot lanes as indicated by the reviewer. We
acknowledge that this was not immediately clear and have clarified it in the figure
legend. Regarding the visual inconsistency between Fig. 1b and 1c, we now **show more**
**representative images** of our results in **Fig 1b**. In addition, we have **re-quantified** all of
our gel images with a second software (www.gelanalyzer.com), which confirmed the
findings initially plotted in Fig. 1c.

	IMAGE J			GELANALYZER		
	AVG	SE	N	AVG	SE	N
SMN1 n.m.	89.9	0.6	5	100.0	0.0	5
SMN2 n.m.	16.7	2.8	8	17.3	3.8	8
SMN2 2C	10.9	1.0	8	0.0	0.0	8
SMN2 3C	54.7	5.2	8	58.9	7.6	8
SMN2 4G	21.5	1.5	8	24.8	1.7	8
SMN2 6C	62.6	4.5	8	60.8	4.3	8
SMN2 9C	47.6	6.7	8	50.1	4.9	8

**2- In Fig 1e the meaning of “others” should be explained in the figure legend**

RNA sequences that fold into hairpins possess the intrinsic potential to form duplexes
given their self-complementarity. In Fig. 1e, RNA samples were folded by snap cooling,
which favors hairpin formation over duplex interactions. Using non-denaturing PAGE we
could confirm the presence of the TSL2 hairpin (lower band) together with a second
conformation (upper band; previously named as “other”), which is consistent with the
presence of duplexes, based on the following:

(1) comparison with the expected band size of the RNA ladder. The **ladder** has now
been included in **Fig. 1e** for clarity.

(2) denaturing electrophoresis conditions using Urea (see **new Supp. Fig. S2**) confirmed
that the two bands observed under native conditions represent folding states, and not
contaminants from the RNA synthesis.

(3) TSL2 duplexes could be observed by NMR under experimental conditions known to
favor duplex formation, including high RNA concentration, high salt content, or high
temperature¹ (not shown). When increasing RNA and salt concentrations, the upper
band from our native gels also increased (see **new Supp. Fig. S2**).

We have modified Fig. 1e and its legend to include this information, as well as generated
the **new Supp. Fig. S2**.

**3- Page 5 lines 112-114 The authors interpret the effect of the 4G mutation in**
**structural terms, they should discuss also the possibility of creating a splicing**
**factor binding site that in turn inhibits splicing of exon 7 independently of the**
**structure. The 4G mutation seems to expose a sequence GUAAGGAGU that**
**although not fully “canonical” could be a binding site of hnRNPA1 or other**
**proteins of that family whose binding will in turn hamper 5'ss recognition**

In a previous report², nearly no increase in *SMN2* E7 splicing was observed upon
mutating residue 16G, which is the base pair of 4C. This supports a conformational
interpretation on the low splicing impact of modifying this region of the hairpin, rather
than a primary sequence effect by mutation 4G. Moreover, the canonical binding
sequence of hnRNPA1 has been well defined as UAGGGA/U³, which differs slightly from
the sequence generated by the 4G mutation. However, we agree with the reviewer in

that we cannot completely rule out that mutation 4G triggers binding of hnRNPA1 or
other inhibitory splicing factors. We have, therefore, incorporated this consideration into
the manuscript:

*“The 4G mutation triggered the strongest conformational changes in TSL2. However,*
*SMN2 splicing was only mildly affected, suggesting that a certain level of TSL2 structure*
*is required for exon inclusion, or that the binding sequence of a splicing factor may have*
*been affected by this mutation.”*

**4- Minor point, there is a typo in Page 7 line 148 TSL2-interacting**

This typo has now been corrected.

**6- Fig 4a and Page8 lines 182-187: It is surprising that with the almost total**
**restoration of normal exon 7 inclusion at the RNA level (Fig 3b and 3c) there is**
**such a marginal increase in protein as seen by western blot. The recovery of the**
**gems (Fig 4b) is certainly impressive but a measure of the total amount of mRNA**
**containing exon 7 before and after treatment with PK4C9 is worth doing. The**
**qPCR tends to give overestimates of the functional intact mRNA present in the**
**cell, a Northern blot will give a more definitive result of the mRNAs including and**
**excluding exon 7.**

We completely understand that a 1.5-fold increase in SMN protein levels (Western blot)
might seem insufficient compared to the total correction of *SMN2* E7 splicing. However,

this is quite commonly seen in the SMA literature. To the best of our knowledge, more
 than a 2-fold increase in SMN protein has not been reported for a small molecule
 modifier of *SMN2* splicing, unless such molecule also increases *SMN2* expression levels
 by activating transcription (*f.e.*, Valproic Acid, VPA⁴; see **Table below**). This can be
 explained because the amount of protein that a splicing modifier can induce is limited by
 the number of *SMN2* mRNA copies present in the cell. A 2-fold increase in SMN protein,
 however, has been shown to (1) be sufficient to reverse SMA phenotypes in mice
 models, including life span and motor function⁹, (2) be the difference between the
 GM03813C fibroblast line (SMA type I, the most severe type of SMA) and the
 GM03814B line (a phenotypically unaffected individual) (see Fig. 4a of our study), (3) is
 the value range of some of the small molecules that have recently reached clinical trials
 for SMA (*f.e.*, trials NCT02268552 and NCT03032172).

Examples of small molecules known to change SMN2 E7 splicing and SMN protein levels				
Molecule	E7 splicing fold (PCR)	Protein fold (WB)	Increases SMN2 transcription?	Reference
PK4C9	1.9 (semi-quantitative PCR) 3.0 (qPCR)	1.5	No	(our study)
C1	1.9	1.7	No	9
C2	1.9	1.5	No	9
C3	1.7	1.5	No	9
NVS-SM1	~15	1.6	No	10
NVS-SM2	~2	1.6	No	10
Hydroxyurea	≤3	≤1.94	No	11
VPA	1.8–5.2	1.8-4.2	Yes	4

qRT-PCR has been used as the gold standard to evaluate *SMN2* E7 inclusion, and the
 primers and Taqman probe used in our study have been validated by us and others⁹.
 qPCR has higher sensitivity and reliability over a greater dynamic range of RNA
 concentrations than other techniques, including semi-quantitative PCR or Northern blot.
 In fact, we have found in the past that *SMN2* expression levels are too low to be
 detected by conventional Northern blot protocols, unless enhanced detection with
 radioactivity is used (to which we do not have access).

Based on all this, we hope that the reviewer will agree with us that a Northern blot would
 bring little added value to the manuscript.

**7- It is not clear how the authors differentiate PK4C9 effects on splicing due to**
 **SMN recovery or to a direct off target effect of PK4C9. The eventual off target**
 **effects seen in WT fibroblasts after PK4C9 treatment should be more widely**

**discussed particularly on the light of potential use as a therapeutic agent.**

Our RNA sequencing (RNA-seq) analysis detected 201 differentially spliced genes with
an absolute PSI (percent spliced in) >0.4 upon treatment of human SMA fibroblasts
(GM03813C) with PK4C9 (40 μ M, 24 h). A series of studies in mice have previously
shown that reduction of SMN protein results in widespread splicing abnormalities, the
identity of which depends on the genetic model, experimental conditions, and tissue/cell
lines used. For example, the following numbers of dysregulated splicing events have
been reported in SMN-depleted mice cells: 145 (motor neurons)¹²; 252 (spinal cord,
post-symptomatic stage), 16 (spinal cord, pre-symptomatic stage)¹³; 104 (motor
neurons), 86 (white matter)¹⁴; 259 (spinal cord), 73 (brain), and 633 (kidney)¹⁵. It is
therefore not surprising that the recovery of SMN protein induced by PK4C9 in SMA
fibroblasts is coupled with a large number of splicing changes, which could represent the
reversal of at least part of such generalized splicing abnormalities and be of therapeutic
relevance. ~25% of the changes found in our RNAseq study affect genes altered in
previous reports in mice. However, a formal comparison between ours and these
previous results has not been conducted in our study, given that the identity of specific
exons and introns affected in SMN-depleted mouse nerve cells has been shown to not
translate to human SMA fibroblasts¹².

Besides PK4C9, there are only two other examples of *SMN2*-splicing modifying small
molecules in the literature for which RNA-seq data also exist, the chemical scaffolds of
which differ notably from PK4C9. In particular:

(1) Novartis: NVS-SM1 (100 nM). 35 differentially spliced genes with PSI>0.4 were
identified¹⁰.

(2) Hoffmann-La Roche: SMN-C3 (500 nM). 13 differentially spliced genes with PSI>0.4
were identified⁹.

In these two cases, the molecules tested were not direct hits from a chemical screen
(like PK4C9), but chemically optimized leads with maximized cellular potency (nM range)
and oral availability. A fair comparison of these two molecules with PK4C9 can therefore
not be made, since PK4C9 is still in the pre-optimization stage. However, we did find
three differentially spliced genes (*SMN2*, *SLC25A17*, and *VPS29*) in common between
the three studies (see **Venn Diagram** below), further supporting that at least some of all

PK4C9-induced splicing changes represent a positive consequence of SMN protein
rescue.

Venn diagram. Genes where alternative splicing events were detected with an absolute PSI of at least 0.4 between treated and control samples. There are three genes that were affected by all three compounds. <http://bioinfogp.cnb.csic.es/tools/venny/index.html>. We would be happy to include this figure in the manuscript should the reviewer agree.

However, part of the PK4C9-sensitive splicing changes are also likely to be off-targets.
In this regard, it is important to keep in mind that PK4C9, in its current state, is not
intended as a therapeutic agent, but as a proof-of-concept molecule that will undergo
chemical optimization to become a more potent and specific lead compound. Being able
to discern between undesired PK4C9-induced off-target vs. SMN recovery-mediated
splicing changes is key for the chemical optimization of PK4C9's specificity. In an initial
low-scale attempt, we compared the effect of PK4C9 on eight of these genes in SMA vs.
WT fibroblasts. To do this, we assumed that (1) true off-targets would be similarly
affected by PK4C9 in WT and SMA cells, but that (2) SMN-dependent changes would
respond differently to treatment in WT vs. SMA cells, given their different SMN starting
levels (see Fig. 4d). Four out of these eight genes belonged to the first case and the
remaining four to the second, confirming the co-existence of both effects. We now plan
to expand this analysis to the rest of transcripts and to combine this information with our
structural results (see Fig. 5, Fig. 6 and Fig. 7), in order to lead the optimization of
PK4C9's specificity.

Finally, a number of less-active, **structural analogues of PK4C9** that do not affect
SMN2 E7 inclusion, and which were synthesized for the revised version of this
manuscript (see **new Supp. Fig. S8**), also failed to modify the splicing of two of the
transcripts that we classified as SMN-recovery dependent, further validating our
conclusions.

**Mentions to all these points** have now been added to the Results and Discussion
sections of the revised version of our manuscript.

**Reviewer #2**

We thank the reviewer for their time and constructive comments. We have performed
additional experiments and text modifications to address their concerns, as detailed
below. To aid reading of the revised manuscript, all major changes have been
highlighted in yellow.

**Remarks to the Author:**

In this manuscript Garcia-Lopez et al. identified a natural compound binding to the stem-
loop RNA structure of SMN2 exon 7 and report a detailed characterization of the mode
of action of this compound *in vitro*, in cells, and in drosophila. The manuscript contains a
large amount of data, is technically sound, and a piece of work of high quality.

However, there are several points that need to be addressed before the manuscript is
suitable for publication.

**1. This regards in particular the fact that the compound binds to the terminal ends**
**of the stem loop structure. In the *in vivo* situation the terminal ends are not freely**
**available. Is the interaction mode determined by NMR and MD compatible with this**
***in vivo* situation?**

To the best of our knowledge, the only techniques that allow for probing of RNA
structure *in vivo* include the recently developed FragSeq, SHAPE-Map, DMS-MaPseq,
and their variants. All these methods are based on enzymatic (FragSeq¹⁶) or chemical
(SHAPE¹⁷, DMS¹⁸) modification in living cells of RNA residues when these are in a single
strand environment, coupled to deep sequencing to detect such modifications.
Unfortunately, these are complex techniques currently used by only a few laboratories,
and that are beyond the expertise of the authors of this manuscript.

The *in vitro* version of SHAPE, however, has been previously performed on a synthetic
RNA with the SMN2 E7/I7 junction sequence¹⁹. The proposed secondary structure of
TSL2 from this study is consistent with our NMR and MD findings (see Figure below).

Finally, to further address the concern of the reviewer, we have used NMR to investigate
 an **extended version of TSL2** (TSL2 23mer) that better mimics its *in vivo* context. In
 particular, the TSL2 23mer contains the sequence of TSL2 (TSL2 19mer) flanked by two
 residues on each side from the endogenous *SMN2* sequence (5'-AC-19mer-AA-3'). The
 23mer sequence forms essentially the same hairpin as the 19mer, as visible by
 comparison of the respective **1D spectra** (see Figure below); and so are the interactions
 with PK4C9, as measured by **WaterLOGSY** spectra (see Figure for point 5 of this
 reviewer, page 13).

**TSL2 19mer vs. 23mer**. The 1D spectra of TSL2 19mer and TSL2 23mer show that the RNA
 hairpin is being formed in both cases. Figures show 1D proton spectra of the imino and the
 aromatic resonance region. For the aromatic resonances, the excitation sculpting scheme was
 employed to suppress the water resonance and for the iminos, the jump-return echo sequence was
 employed. We would be happy to include this figure in the manuscript should the reviewer agree.

**2. From figure 6 d and 6 there seem to be only a few contacts between the natural**

**compound and the RNA. Wouldn't the authors expect a broad range of unspecific**
**effects?**

We understand the concern of the reviewer. However, there are a number of examples
of small molecules that interact selectively with RNA through a low number of hydrogen
bond contacts combined with a significant hydrophobic contribution. For example, the
small molecule inhibitor of viral replication DB213 bound to the HIV-1 frameshift site
RNA (PDB code: 2L94); the RBT550 inhibitor of the HIV-1 TAR RNA, where the indole
moiety interacts with its target exclusively via hydrophobic contacts and π - π stacking
(PDB code: 1UTS); or natural product Theophylline, which also binds to an aptamer
RNA through hydrophobic and π - π stacking interactions (PDB code: 1O15).
Aminoglycosides, a class of antibiotics, also bind to ribosomal RNA only via electrostatic
interactions²⁰. In our previous version of the manuscript, 40-ns MD simulations identified
a few but significant hydrogen bond contacts between PK4C9 and TSL2 residues U2
and U19, as well as π - π stacking with G18 and hydrophobic interactions with A1 and
U3. In our revised version, this **analysis has been extended** to 100 ns, in order to
increase the statistical significance of our findings. The extended simulations
corroborated the combined contribution of hydrogen bonds, π - π stacking, and
hydrophobic contacts to the binding mode of PK4C9 (see representative conformations
in the figure below and revised **Fig. 6**). To allow for a more detailed visualization of these
interactions, two Supplementary Videos (**Supp. Vid. V1** and **Supp. Vid. V2**) from our
MD simulations have been included to the revised manuscript.

2D view of the interaction between TSL2 and PK4C9. Representative structures from the most populated clusters of the 40-ns (top) and extended (100-ns, bottom) MD trajectories. Similar interactions were found for both trajectory lengths. Interactions were plotted using the Maestro software. Note that these images show static snapshots structures. Slight dynamic changes in the interactions can occur during the trajectory.

We do, however, acknowledge the presence of non-specific, off-target effects of PK4C9,
as already discussed and observed in our RNA-seq experiment. This aspect will be
addressed during the chemical optimization phase of PK4C9 to convert this hit molecule
into a more potent and specific lead compound, taking advantage of our acquired
understanding on how PK4C9 interacts with its target.

**A more detailed discussion** of these considerations has been included in the revised
version of the manuscript.

**3. The NMR chemical shift perturbations shown in Figure 5d seem rather small to**
**me given that the compound has aromatic rings. Wouldn't the authors expect**
**larger chemical shift perturbations due to ring current effects.**

Ring current effects or binding effects would be best observed on the U2 imino proton
and the G18 imino proton. Unfortunately, G18 and U2 imino protons are generally
invisible by NMR due to fast water exchange and are therefore poor reporters. In
addition, when adding PK4C9 to the RNA the presence of DMSO does not allow for
reducing temperature, which would reduce exchange.

**4. Do the chemical shifts back-calculated from the MD model match what they see**
**in the spectrum?**

Chemical shift prediction based on our MD models was not attempted due to two main
reasons. (1) Whilst a plethora of empirical structure-based chemical shift predictors have
been developed for proteins²¹⁻²⁶, unfortunately only a few such methods exist for RNA
(f.e., NUCHEMICS²⁷ or SHIFTS²⁸). These programs can predict nonexchangeable ¹H
chemical shifts, but are generally not precise enough for imino protons. (2) The relatively
short time scale of our MD simulations (100 ns) is not sufficient for reliable back-
calculation of certain parameters that have been reported to require simulation times >1
μ s (f.e., J-couplings across hydrogen bonds)²⁹. However, a number of observations
cross-validate our NMR and MD structural results. For example, (1) both NMR and MD
identified the same TSL2 residues (*i.e.*, U2 and U19) as mediators of PK4C9 binding; (2)
NMR and MD found a similar proportion of triloop and pentaloop conformations in the
TSL2 ensemble; and (3) our combination of NMR and MD lead to predictions on the
effect on E7 splicing of the 2A, 2C, 3G17C and 8G12C TSL2 mutations that could be

validated experimentally in human cells (see Fig. 7).

We have now **revised the text** of our manuscript to include clarifications to the points 3
and 4 of the reviewer.

**5. Did the authors follow the chemical shift changes of the ligand by NMR and**
**could they validate their MD data with respect to the ligand - stem-loop RNA**
**interactions shown in Figure 5?**

PK4C9 showed very small CSPs upon titration with RNA excess, which are uniform (see
Figure below). To better observe weak interactions, we performed **WaterLOGSY**
(Water-Ligand Observed via Gradient SpectroscopY) experiments, a method commonly
used for primary NMR screening in the identification of compounds binding to the target
of interest in the μM range³⁰. These experiments showed negative NOEs (*i. e.*,
magnetization transferred from “bound water”) for the ligand in the presence of TSL2
19mer and 23mer, thus confirming binding of PK4C9 to TSL2 also from the ligand’s side.

**PK4C9 binding to TSL2.** (a) Spectrum showing CSPs on PK4C9 upon addition of 2 equivalents of TSL2. The small
CSP size is likely due to low binding affinity. To overcome this, WaterLOGSY was conducted (b), which detected
negative NOEs, thus confirming binding. Blue: 1D reference spectrum of TSL2 plus 10-fold excess PK4C9 in 80%
NMR buffer, 20% DMSO-d₆; red: WaterLOGSY of PK4C9 with TSL2 19mer; green: WaterLOGSY of PK4C9 with TSL2
23mer. We would be happy to include this figure in the manuscript should the reviewer agree.

**6. Minor points:** These errors have now been corrected.

- **some figure labels are too small and hard to read, e.g. Fig. 1a numbering of**
**nucleotides etc.**

- **some figures miss proper axes labels/units, e.g. Figure 1d (what is 'max'), Figure**
**1f (CD)**

In this particular case, we would like to note that Mol CD (Molecular Circular Dichroism)
are standard Circular Dichroism units.

- **in some figures a broken axis is shown, but there are no data points extending to**
**these values (e.g. Fig. 1e, Fig. 1d)**

- **some figures miss error bars, e.g. Fig. 1d,**

In this particular case, we would like to note that the standard error bars in Fig. 1d and
1e have been plotted, but they are too small to be visible.

- **Figure 3e 'Drosophila MN' is there anything missing?**

- **Figure 4 seems of poor resolution**

**Reviewer #3**

We thank the reviewer for their time and constructive comments. We have performed
additional experiments and text modifications to address their concerns, as detailed
below. To aid reading of the revised manuscript, all major changes have been
highlighted in yellow.

**Remarks to the Author:**

Garcia-Lopez et al show that one can use drugs (PK4C9 as most efficient) to rescue the
reduction of SMN1 by mis-splicing, causing SMA, by induction of a secondary structure
change of the isoform SMN2 becoming splicing competent at the required point. In this
paper many methods are coherently represented and a comprehensive evidence based
conclusion is drawn.

Most methods are standard in the field, as is most of the application. However, the
detailed structural work for finding out the mechanism of drug action is unusual and
brings out new information about this drug and the splicing mechanism - at least partial -
some of this was already suggested at lower resolution. This work is well done and
should set a standard for the field of drug development on RNA drugs - to include more
detailed mechanistic studies with the completeness of methods.

**Major points should be clarified:**

**1a. Line 275-277: Why is this coupling assumed? (loop formation to terminal**
**opening)**

We apologize for not having explained this conclusion more clearly. The coupling
between loop closing and terminal opening - which one could envision as a clothes peg,
where tightening from one end makes the other end open - is assumed given that the
distance between residues A1-U19 increases in the presence of PK4C9, whereas the
distance between residues A8-U12 (closing pair of the loop) decreases, even though the
ligand does not directly bind to this region. This is generally observed in the induced fit
model for ligand recognition, by which ligands may induce a conformational change in
the target rather than selecting a conformation from a pre-existing population.

**1b. ...how is this interaction/conformational change information transported over**

**that long distance?**

By inducing a terminal opening of the TSL2 stem the overall structure of the hairpin has
to readjust in order to stay energetically stable. Tightening of the loop would be a way of
achieving this, as it would increase the energy levels in that region, and compensate for
the energy release at the other end of the hairpin. Similar phenomena have been
previously described, including the example of adenylate kinase³¹, where upon substrate
binding, the enzyme increases its chain mobility in a region remote from the active
center. This region 'solidifies' again upon substrate release, serving as a 'counterweight'
balancing the substrate binding energy. A **reference to this example** has now been
included in the manuscript text.

**2. Could some of it simply be explained by less overall stability (e.g. U2A**
**mutation) -> add melting temperatures (e.g. from Fig S2, but preferred by UV),**

We agree with the reviewer in that TSL2 mutations and PK4C9 most likely affect the
overall stability of the RNA. However, simply reducing TSL2 stability in a “blind” way
would not necessarily aid 5' ss recognition. For example, mutation 8G12C, which has
the highest **melting temperature** (T_m; calculated using differential scanning fluorimetry,
DSF) amongst all thirteen TSL2 mutations studied (see Table & Figure below and **new**
**Supp. Table S1**), showed the best *SMN2* E7 inclusion values in HeLa cells transfected
with mutated minigenes. In fact, we did not find a significant correlation between the T_m
of the thirteen mutant RNAs and their impact on *SMN2* E7 splicing (see Figure below),
further arguing against a general stability effect on TSL2 as the reason for E7 inclusion.
Instead, the different mutations would trigger conformational rearrangements that may
directly or indirectly improve accessibility of the 5' ss. This idea is in agreement with our
2-amino purine (2AP), Circular Dichroism (CD), and MD results (Fig. 1 & 6, **Supp. Table**
**S1**), and with the fact that mutation 8G12C can increase 5' ss accessibility despite being
located at the other end of the stem.

T _m values of TSL2 mutants obtained by differential scanning fluorimetry (DSF)													
	4G	2G	9C	3C	WT	17C	6C	3G	3A	3G17C	2A	2C	8G12C
T _m	41.2	42.0	44.7	45.5	45.6	47.6	48.7	49.2	51.1	51.0	51.5	51.8	52.2
± SE	±0.16	±0.02	±0.05	±0.00	±0.07	±0.03	±0.09	±0.05	±0.05	±0.05	±0.07	±0.06	±0.02

**Representative DSF curves.** Upon request by the reviewer T_m values were obtained by DSF, as described in³² (n=8,
 Boltzmann sigmoidal fitting). A plot showing no correlation between T_m values and SMN2 E7 inclusion is included
 (right). UV melting curves could not be obtained due to lack of appropriate equipment. We would be happy to include
 this figure in the manuscript should the reviewer agree.

**3. Line 129-132: would that not also decrease specificity?**

The reviewer raises the concern that the screening of focused chemical libraries may
 decrease hit specificity. On the contrary, the use of focused libraries has increased
 notoriously in recent years, due to a number of benefits, as reviewed in refs.^{33,34}. For
 example, commercially available, high-chemical diversity libraries are generally biased
 for modulating protein function, thus yielding much lower hit rates for RNA targets (0% to
 1%, ref.³⁵ plus our own experience and personal communications). In addition, often the
 hits identified are not specific for the RNA probed and are likely to have protein off-
 targets³⁶. In contrast, a well-designed, target-focused library with privileged scaffolds to
 bind RNA can (1) save time and money by reducing the number of compounds to be
 experimentally tested, (2) yield higher hit rates by eliminating compounds that are
 unlikely to bind to the target^{35,37}, and (3) find more potent and selective binders, as it has
 been shown for inhibitors of the c-Src kinase³⁸.

It is of course a possibility that hits from a focused library of RNA binders bind to more
 than one RNA target, which is a common concern to all chemical screening campaigns.
 Typically, these campaigns are followed by a chemical optimization phase of the
 identified hits into clinical candidates, where specificity is closely monitored throughout
 the process. In this regard, having performed a target-based screening poses a big
 advantage vs. phenotypic screening approaches, since our knowledge on the
 mechanism of action of PK4C9, coupled with our RNA-seq findings, will accelerate the
 optimization of PK4C9's specificity. It is also worth noting that focused collections often
 offer molecular starting-points that dramatically reduce the subsequent hit-to-lead
 optimization timescale, given that the properties of their compounds have already been

filtered to suit the type of target in question.

A **mention to focused library screening** has been added to the Discussion of the
revised manuscript.

**4. Some minor changes need to be addressed:**

**a. line 114/115: what indicates the native page conclusion of hairpin formation? –**
**more detail please (e.g. marker...)**

RNA sequences that fold into hairpins possess the intrinsic potential to form duplexes
given their self-complementarity. In Fig. 1e, RNA samples were folded by snap cooling,
which favors hairpin formation over duplex interactions. Using non-denaturing PAGE we
could confirm the presence of the TSL2 hairpin (lower band) together with a second
conformation (upper band; previously named as “other”), which is consistent with the
presence of duplexes, based on the following:

(1) comparison with the expected band size of the RNA ladder. The **ladder** has now
been included in **Fig. 1e** for clarity.

(2) denaturing electrophoresis conditions using Urea (see **new Supp. Fig. S2**) confirmed
that the two bands observed under native conditions represent folding states, and not
contaminants from the RNA synthesis.

(3) TSL2 duplexes could be observed by NMR under experimental conditions known to
favor duplex formation, including high RNA concentration, high salt content, or high
temperature¹ (not shown). When increasing RNA and salt concentrations, the upper
band from our native gels also increased (see **new Supp. Fig. S2**).

We have now modified Fig. 1e and its legend to include this information, as well as
generated the **new Supp. Fig. S2**.

**4b. CD experiments: melting curves: maybe the effect is caused by difference in**
**stability rather than conformation?**

Conformational and stability changes are linked – it is through conformational changes
that the overall thermodynamic stability of a structure is affected. In the manuscript, we
chose to use the expression “conformational changes” rather than “stability changes” to
avoid confusion. For example, a mutation can stabilize (*i.e.*, freeze) the RNA in a
particular conformation that makes the overall structure less stable (*i.e.*, lower melting

temperature, T_m) - this mutation could therefore be considered as stabilizing or
destabilizing depending on which of the two things are being discussed. Assuming that
the reviewer means stability in terms of T_m, we refer to our **new Supp. Table S1**; where
a summary of our 2-amino purine, circular dichroism, and native PAGE results clearly
shows that the different mutations induce conformational changes, the magnitude of
which correlate with E7 inclusion but not necessarily with T_m values (see also Fig. 1).

**4c. Suppl. Table S1 is wrongly called Suppl. Table S2, Tables S3,4 & 6 are missing**
**or wrongly assigned -> Supplementary tables are a mess**

We apologize for not having organized the Supporting Material more clearly. Supp.
Tables S1, S3 & S4 (now renamed as S2, S4 & S5) are large Excel files that were
provided as separate files, whereas Supp. Tables S2 & S5 (now renamed as S3 & S6)
were part of the Supporting Material Word document. To avoid this confusion, the new
Supporting Material Word document now contains the list and captions of all Supp.
Tables and a reference to the corresponding Excel file.

**4d. Line 163 & 166: please define what control cells represent (cells with SMN1**
**basal mRNA levels? Or just the same cells – DMSO influence – as mentioned in**
**line 194?) -> for immunohistochemistry it is mentioned what control is as cells –**
**not for drug testing (line 433) and for Figure 5 (line 640)**

We have now clarified this as follows:

- Lines 163 & 166: controls cells are now called DMSO-treated cells. These are the
same cells as those treated with PK4C9, but treated with DMSO under otherwise the
same conditions.

- Line 433: our previous Methods description read: “*At 80 % confluence, cells were*
*treated for 24 h with DMSO (control) or PK4C9 (40 μM)*”. This now reads: “*At 80 %*
*confluence, cells from the same cell line were treated for 24 h with DMSO (control) or*
*PK4C9 (40 μM)*”

- Figure 5a: our previous figure legend read: “[...] *AGGTAAG as the most enriched motif*
*in exons that are differentially spliced in SMA cells upon treatment with PK4C9 (40 μM,*
*24 h)*”. This now reads “[...] *AGGTAAG as the most enriched motif in exons that are*
*differentially spliced in SMA cells upon treatment with PK4C9 (40 μM, 24 h) compared to*
*SMA cells treated with DMSO (24 h).*”

- Figure 5b-c, our previous figure legend read: "*Transfected cells were treated with*
*DMSO (0.04%, controls) or PK4C9 (40 μM) (n>3)*". This now reads: "*Transfected HeLa*
*cells were treated with DMSO (0.04%, controls) or PK4C9 (40 μM) (n>3)*".

**4e. Fig 5d: CSP are significantly different between DMSO and PK4C9, however,**
**they have the same direction, indicating the same underlying structural change**
**(potentially in this case destabilization)? – please explain -> U11 different trend**

We understand the reviewer's comment. However, CSPs and their direction are often
easy to over-interpret. A same trend could indeed mean the same kind of structural
change, but not necessarily. The chemical shift resonance of an atom is not only
affected by the local structural geometry, but also influenced by changes in the
environment (such as ligand binding, solvent, electro-negativity of nearby groups,
induced magnetic field effects, etc). It is, therefore, important to interpret these data
taking into consideration their context. In our case, the PK4C9-induced CSPs observed
by NMR match the contacts predicted by our non-biased MD protocol, and these two
things together also explain other experimental observations (*f.e.*, the behavior of TSL2
mutants, or of our PK4C9 structural analogues). Regarding U11, we have removed our
previous interpretation of the PK4C9-induced CSP for this residue, as we agree it is less
clear than the cases of U2 and U19.

**4f. Fig 6g: U19 data missing, hence line 273 not possible to observe (U19 also**
**missing in 6C)**

This mistake has been corrected. Graphs from Figs. 6c and 6g now include residue U19.

**4g. Please make an overview table over mutants and effects (E7 inclusion,**
**terminal opening, tri/penta-loop distribution, PK4C9 activity, stability)**

This is an excellent idea, which we have now incorporated to our manuscript as **Supp.**
**Table S1.**

The following mistakes have also been corrected:

- **Line 148 TSL-2 interacting**

- **Line 376 "one of the few examples ..."**

- **Suppl. line 280, comparisson (with just one s)**

**Reviewer #4:**

We thank the reviewer for their time and constructive comments. We have performed
additional experiments and text modifications to address their concerns, as detailed
below. To aid reading of the revised manuscript, all major changes have been
highlighted in yellow.

**Remarks to the Author:**

Garcia-Lopez et al describe the identification of a novel small molecule binder of the
TSL2, RNA stem loop structure in the SMN2 gene. Using a combination of NMR, Next
generation sequencing and mutagenesis studies they go on to show that PK4C9 binds
to TSL2, promoting a conformational shift that favors increased inclusion of SMN2 exon
7.

**The current study represents the first study describing the identification of a small**
**molecule modulator of TSL2 but the importance of TSL2 as a key element in**
**regulating SMN2 splicing has been previously well documented (Singh et al; NAR,**
**2007) and is recognized as an attractive target in the field. Additionally, the idea of**
**using small molecules to target RNA secondary structure has attracted a lot of**
**attention and has clear precedent (e.g Velagapudi et al; NCB 2013; Childs-Disney**
**et al; ACS Chem Biol 2014; Velagapudi et al; PNAS, 2016; Patwardhan et al,**
**MedChemComm 2017). Hence, the current study does not offer any significant**
**advancement in our understanding of achieving SMN2 splicing modulation.**

It is unfortunate that the reviewer finds our study of little novelty and we apologize if the
discrepancy arises, at least partly, due to poor explanation from our side. However, we
are convinced that our work contributes with new relevant aspects and tools to the SMA,
RNA splicing & drug discovery fields, and would like to emphasize some of the key novel
contributions of our study. In particular:

1. TSL2 was first described in 2007 by *in vitro* enzymatic probing, followed by *in vitro*
SHAPE in 2015^{2,19}, but no atomistic characterization of this structure has ever been
conducted. In our study, we have **solved the first atomistic structure of TSL2** using
NMR. This NMR structure has been submitted to the PDB and is now available to any

laboratory who wishes to conduct structural research on TSL2, both experimentally and
*in silico* (f.e., drug design, RNA-protein interaction modeling, etc). Thus, our work
provides a highly valuable tool that will make certain experimental designs possible for
the first time to different research communities.

2. The report from 2007 described TSL2 as a triloop, whereas the SHAPE study from
2015 found it in a pentaloop form. Here, we **resolve a discrepancy** and describe that
both species do co-exist. In fact, we provide the most extensive description of the
conformational dynamics of TSL2 (both wild type and mutated) to date using MD for the
first time, along with the first rational and atomistic explanation as to how the TSL2
equilibrium between pentaloop and triloop conformations could influence E7 splicing.
This information can certainly inspire other groups interested in the study of *SMN2*
splicing regulation, and/or in the study of stem-loops regulating splice site recognition.
As an example, see reference³⁹ - where terminal loop stability was predicted to affect
splice site recognition.

3. We have performed the **first target-based screening** of small molecules for SMA.
Several excellent small molecule phenotypic screening campaigns have been
undertaken for SMA, but these relied on splicing reporters in living cells, and very often
the mechanisms by which hits change splicing remained unsolved. This is not the case
in our study. Moreover, our target-based screening was conducted on an RNA target. As
the reviewer very correctly points out, ours is not the first use of small molecules to
target an RNA structure^{36,40,41}. However, it is known that targeting RNA remains a
challenging field with still few precedents compared to proteins targets. Thus, our study
will contribute to the expansion of this field by providing an additional and novel
approach. For example, our study has generated all the necessary structural tools and
knowledge for other groups to attempt *in silico* drug design against TSL2. To the best of
our knowledge, *in silico* drug design has never been attempted for SMA.

4. Experimental target-based screenings and *in silico* rational drug design against RNA
structures have been attempted for Myotonic Dystrophy⁴²⁻⁴⁴, cancer⁴⁵, the HIV TAR
RNA^{46,47} or different miRNAs^{48,49}, among others. However, when this project started,

there were no clear precedents of bioactive small molecules targeting the RNA structure
of a splice site. Since then, two examples were published in 2014 of rational screenings
targeting the stem loop of the E10/I10 junction of *MAPT*^{50,51}. This rather limited number
of precedents demonstrates that targeting RNA structures with small molecules to
modulate splicing is indeed a **promising field still in growth**, which will benefit from
additional examples.

5. PK4C9, our most promising hit from a proof-of-concept screening, offers a **new**
**chemical scaffold** in the SMA literature. We are aware that PK4C9 is not a final drug
candidate, but a starting point for chemical optimization. This optimization has the
potential of identifying an entirely new series of active compounds for SMA, which we
are committed to continue developing. The fact that PK4C9 is not in its final state has
the added potential of raising interest in the field by other groups that wish to contribute
to its development.

We have revised our manuscript text to include mentions to these points.

**1. While the authors performed a screen for TSL2 interactors and identified their**
**top hit PK4C9 using this approach, they fail to provide any compelling evidence**
**for the selectivity of their hits. It is critical that they counter-screen their hits on**
**unrelated, control RNA secondary structures to provide evidence that PK4C9 and**
**related hits do not act as promiscuous RNA binders.**

The reviewer raises an important point. Whilst we do not expect complete target
specificity at the current stage of development of PK4C9 (as already discussed and
observed in our RNAseq experiment), we agree with the reviewer in that it is crucial to
confirm that PK4C9 is not a pan-nucleic acid binder. To address this issue, we have
performed a **new experiment** using fluorescence displacement with RiboGreen (see
**new Supp. Fig. S6**), where binding of our top screening hit compounds to native TSL2
was compared with the following controls of binding selectivity:

1. An unrelated RNA structure. Here, NCI377363, PK4C9, and BJJ466 showed a
significantly better interaction with native TSL2 than with the control structure (see figure
below), indicating some binding selectivity. 288D, however, showed weak binding to
both.

2. A partially denatured TSL2 (60 °C). A dramatic decrease in NCI377363, PK4C9 and
BJGF466 binding was observed when TSL2 was, at least partially, unfolded (Tm of
TSL2 under the assay conditions is 45.6 °C), indicating that these hits target TSL2
structure rather than its primary sequence, and that they do not efficiently bind to single-
strand RNA. Conversely, 288D binding was not affected.

Collectively, these results demonstrate that NCI377363, PK4C9 and BJGF466 are not
promiscuous RNA binders, whereas 288D seems to be.

**NCI377363, PK4C9 and BJGF466 are not indiscriminate RNA binders.** (a) Sequence and secondary structures of
TSL2 and the selected unrelated RNA control (RNAfold online tool). (b) RiboGreen fluorescence displacement binding
assay. Upon binding of hit compounds to the RNA, the fluorescence of the dye decreases. Data points show mean
values \pm SE (n=3). RiboGreen can bind to double stranded and to single stranded nucleic acids.

**2a. The binding activity is only demonstrated indirectly using fluorescent**
**techniques, well-known to be prone to false positives, without any counter-screen**
**data. The structural aspect of compound binding is simulated based on RNA NMR**
**data, virtual docking and molecular dynamics. While the authors do show NMR**
**shifts of RNA when compounds are added, these could well be conformational**
**changes in RNA which could be induced by metal ion contaminations in the**
**compound or by metal ion chelation by the compound (the compound does**
**resemble a chelator). Can they rule out this possibility?**

The reviewer raises two interesting possibilities:

1. That traces of metal ion contaminants in the synthetic sample of PK4C9 are
responsible for the TSL2 conformational changes observed by NMR. This possibility can
be safely ruled out, as the only metal ions that the PK4C9 solution contained were traces
from the environment, which we have quantified by **atomic absorption (ICP)** as per the
table below. The only metal that could have originated from the synthesis of PK4C9 is
palladium, which is near the limits of detection (thus truly trace amounts), and which
could not have affected the conformation of TSL2, as this would require a stoichiometric
amount of metal.

ICP analysis of metals in the PK4C9 sample			
Element	w/w (ppm)	e(w/w)	melement (mg)
Pd	5.157370195	10.56570764	2.57869E-05
Mg	137.3612737	38.91884638	0.000686806
Fe	297.0323541	73.1320868	0.001485162
Co	0	8.820226795	0
Ni	0	9.219686127	0
Cu	0	10.68253924	0

2. That a potential chelating effect of PK4C9 is responsible for the conformational
changes in TSL2 observed by NMR. We understand the reviewer's concern due to the
presence of nitrogens in the composition of PK4C9. However, this concern would apply
to all nitrogen-containing inhibitors described in the literature. We would like to point out
that our NMR buffer (10 mM sodium phosphate buffer pH 6.4, 50 mM NaCl and 0.1 mM
EDTA) not only does not contain any metals, but it also contains the chelating agent
EDTA at 0.1 mM (added to protect the RNA sample from heat-induced RNase digestion
during RNA folding). In our ligand titration experiments, PK4C9 is used at 40-50 μ M.
Therefore any potential chelating effect by PK4C9 would be marginal compared to
EDTA. Finally, even if PK4C9 acted as a transporter of metal ions from the medium into
the RNA, PK4C9 would have a kinetic effect only, which may be relevant in cellular
tests, but could not account for our *in vitro* results, as there is no kinetic barrier to metal
diffusion.

In summary, we are confident that the chemical shift perturbations (CSPs) observed in
our ligand titration experiments represent targeted binding of PK4C9 to TSL2. Our MD
simulations (performed in a metal-free simulated environment) and our NMR results are
convergent, and this convergence lead to predictions that could be confirmed

experimentally. Moreover, the same CSPs have always been obtained in every PK4C9
titration replicate that we have performed.

**2b. Can they show shifts in compound spectra as well?**

In TOCSY experiments, PK4C9 showed very small CSPs upon titration with RNA
excess, which were uniform (see Figure below, a). To better observe this type of weak
interactions, we performed **WaterLOGSY** (Water-Ligand Observed via Gradient
Spectroscopy) experiments, a method commonly used for primary NMR screening in the
identification of compounds binding to the target of interest in the μM range³⁰. These
experiments showed negative NOEs (*i. e.*, magnetization transferred from “bound
water”) for the ligand in presence of TSL2 (see Figure below, b), thus confirming binding
of PK4C9 to TSL2 also from the ligand’s side.

**PK4C9 binding to TSL2.** (a) TOCSY spectrum showing CSPs on PK4C9 upon addition of 2 equivalents of TSL2. The
small CSP size is likely due to low binding affinity. To overcome this, WaterLOGSY was conducted (b), which detected
negative NOEs, thus confirming binding. Blue: 1D reference spectrum of TSL2 plus 10-fold excess PK4C9 in 80%
NMR buffer, 20% DMSO-d₆; red: WaterLOGSY of PK4C9 with TSL2 19mer; green: WaterLOGSY of PK4C9 with TSL2
23mer. We would be happy to include this figure in the manuscript should the reviewer agree

**2c. Bottom line is that they show no evidence for direct, selective binding such as**
**SPR and ITC. It is imperative that they provide direct evidence of compound**
**binding to TSL2 using independent, label-free biophysical methods as suggested**
**above.**

In our study we show proof of direct binding obtained by NMR. NMR is an extremely
powerful technique compared to other biophysical methods, due to its higher sensitivity
and atomic resolution, as well as the fact that NMR allows for monitoring of RNA integrity
whereas techniques like SPR or ITC do not. Nevertheless, to address the issue raised
by the reviewer we have undertaken additional efforts to provide further evidence of
direct binding. In particular:

1. In addition to our original fluorescent displacement (FD) assay using the TO-PRO1
dye, we also provide now evidence of PK4C9 binding to TSL2 using fluorescence
displacement with the RiboGreen dye (see Figure below). Although this is a
fluorescence-based technique that does not directly demonstrate binding, it rules out a
false positive caused by the TO-PRO1 dye.

RiboGreen assay. Incubation of TSL2 with the RiboGreen nucleic acid dye in the presence of DMSO or increasing concentrations of PK4C9. When bound to TSL2, RiboGreen fluoresces. When PK4C9 binds to TSL2, the dye is released and fluorescence decreases. Signal is corrected to the blank (RiboGreen plus PK4C9 alone) and is shown normalized to the DMSO control. We would be happy to include this graph in the manuscript should the reviewer agree.

2. In collaboration with Creoptix (www.creoptix.com), we also attempted an **SPR-based**
**method** to detect direct binding, using the Creoptix WAVE Delta instrument instrument,
which measures grating-coupled interferometry signals. In these experiments, a
biotinylated TSL2 RNA was immobilized onto the matrix surface with PK4C9 being in the
mobile phase. A biotinylated DNA served as negative control. PK4C9 gave a
concentration-dependent signal on the tested RNA that was higher than the one
observed for the DNA control (see Figure below). However, we observed a residual
sticky effect of the ligand to the matrix surface, which although partially occurred, also to
a lesser extent, on an empty surface (*i.e.*, when no RNA or DNA is immobilized). Despite
big efforts (including trying different matrices, buffers, detergents, and controls) we could
not solve this technical issue, which was most likely caused by a combination of the
hydrophobic nature and low solubility of PK4C9 together with the need to work at high

ligand concentrations due to a low affinity binding, leading to precipitation of PK4C9 onto
the surface. Taken together, although these results showed direct binding of PK4C9 to
TSL2, SPR was in our case not suitable to draw a firm confirmation due to technical
limitations.

**Grating-coupled interferometry signals.** Surface Plasmon Resonance (SPR)-based method. Signals originated from
the injection of increasing amounts of PK4C9 into the captured Biotinylated TSL2 RNA (blue) or DNA (red). X axis is
time (s) of injection of PK4C9 from 0 to 120 s.

3. Our group has a strong expertise in **isothermal titration calorimetry (ITC)**^{52,53}.
Therefore, ITC was attempted despite some concerns with respect to the low solubility
and affinity of PK4C9, both of which pose important limitations to this methodology. The
ITC experiment showed weak binding enthalpy for an endothermic low affinity interaction
upon ligand titration (see Graph below). The shape of the observed binding isotherm
(lower panel) is reminiscent of an incompletely described binding event (c -value < 1), as
it would be expected for a K_D in the range of 100-200 μM. Indeed, simulating an ITC
titration under our experimental conditions predicted a K_D value of 100 to 250 μM for our
system. Unfortunately, due to the poor solubility of PK4C9 and limited availability of
TSL2 RNA, it is not reasonable to increase their concentrations to a level that would
allow measuring complex formation. Therefore, we are unfortunately limited by the
capacity of ITC to provide a K_D value that would firmly quantify the binding of PK4C9 to
TSL2.

Binding of PK4C9 to TSL2 RNA measured by ITC. TSL2

(57 μM in a 1.45 ml cell) was titrated with a first 2 μl control injection followed by 29 injections of 7 μl of PK4C9 (2.0 mM in syringe) in presence of 20% DMSO. The raw data for consecutive injections of PK4C9 to TSL2 (top panel) was integrated and corrected for the heat of dilution of the corresponding control experiment (PK4C9 into buffer; data not shown) and plotted against the [PK4C9]/[TSL2] ratio (lower panel).

In conclusion, we have done everything we could to address the issue raised by the
 reviewer. Our fluorescence displacement assays, both with TO-PRO-1 and RiboGreen
 dyes, detected binding between PK4C9 and TSL2. We could not firmly quantify the
 strength of the binding neither by an SPR-like method nor by ITC, due to the low affinity
 and solubility of PK4C9. However, both techniques suggest a weak but direct interaction,
 for which no K_D could be calculated but could be estimated to be around 200 μM . NMR,
 which has the highest resolution of all three techniques, could however detected direct
 binding.

**2d. They should also provide an assessment of binding selectivity using control**
 **RNA structures when carrying out these assessments.**

Please, see our answer to point 1 of this reviewer, where we conclude that PK4C9 is not
 a promiscuous RNA binder by comparing the interaction with native TSL2 vs. denatured
 TSL2 and an unrelated RNA secondary structure, using fluorescence displacement (see
 also our **new Supp. Fig. S6**).

**3a. Do the authors have any evidence for dose responsiveness of their compound**
 **in the SMN2 mini-gene or SMN protein assays? The effect in the minigene assay**
 **and in the SMA patient fibroblast splicing assessment (Fig 3) looks like an all or**
 **none response.**

As shown in Figs. 3b and 3c, the *SMN2* splicing modifier effect of PK4C9 follows a dose-
response curve with typical sigmoid shape, which exploits the maximum possible
amplitude of the response. However, we agree with the reviewer in that the
concentration range at which we see this dose-response (from 10 μ M to 50 μ M) could be
seen as rather narrow. This type of response may be caused by the mode of action of
the molecule, involving conformational changes that are thermodynamically costly, thus
resulting in low potency (μ M range). As an example, this thermodynamic penalty has
been observed and characterized in the case of imatinib binding to C-Src⁵⁴. A reference
to this point has now been added to the legend of Fig. 3, which reads as follows:

*“The dose-response curve of PK4C9 in both cell lines reveals a rather narrow*
*concentration window but achieves maximal response. Concentrations higher than 50*
*μ M could not be measured due to poor solubility of the compound under the*
*experimental conditions”.*

**3b. The protein increase in Fig 4a is very modest at the single dose (40uM) for**
**which data is shown.**

We completely understand that a 1.5-fold increase in SMN protein levels (Western blot)
might seem insufficient, especially given the nearly total correction of *SMN2* E7 splicing.
However, this is quite commonly seen in the SMA literature. To the best of our
knowledge, more than a 2-fold increase in SMN protein has not been reported for a
small molecule modifier of *SMN2* splicing, unless such molecule also increases *SMN2*
expression levels by activating transcription (*f.e.*, Valproic Acid, VPA⁴; see Table below).
This can be explained because the amount of protein that a splicing modifier can induce
is limited by the number of *SMN2* mRNA copies present in the cell. A 2-fold increase in
SMN protein, however, has been shown to (1) be sufficient to reverse SMA phenotypes
in mice models, including life span and motor function⁹, (2) be the difference between
the GM03813C fibroblast line (SMA type I, the most severe type of SMA) and the
GM03814B line (a phenotypically unaffected individual) (see Fig. 4a of our study), (3) is
the value range of some of the small molecules that have recently reached clinical trials
for SMA (*f.e.*, trials NCT02268552 and NCT03032172).

Examples of small molecules known to change SMN2 E7 splicing and SMN protein levels
--

Molecule	E7 splicing fold (PCR)	Protein fold (WB)	Increases SMN2 transcription?	Ref.
PK4C9	1.9 (semi-quantitative PCR) 3.0 (qPCR)	1.5	No	(our study)
C1	1.9	1.7	No	9
C2	1.9	1.5	No	9
C3	1.7	1.5	No	9
NVS-SM1	~15	1.6	No	10
NVS-SM2	~2	1.6	No	10
Hydroxyurea	≤3	≤1.94	No	11
VPA	1.8–5.2	1.8-4.2	Yes	4

**3c. In the minigene assay (Fig 3 a) it looks like PK4C9 and BJGF466 elicit maximal**
**exon 7 inclusion at early timepoints and the effect tends to get weaker at the 24 hr**
**time-point. Do the authors have an explanation for this? Have they looked at later**
**time points?**

The effect pointed out by the reviewer was indeed observable in HeLa cells, but not in
fibroblasts, where the PK4C9 splicing modifier activity did increase with time (see Graph
below, not included in the manuscript).

SMN2 E7 splicing upon 6h and 24 h of PK4C9 treatment.
Semi-quantitative PCR results from SMA fibroblasts. Graph shows mean values \pm SE (n=3). We would be happy to include this graph in the manuscript should the reviewer agree.

There are several possible explanations for this
cell-specific difference:

1. Whilst PK4C9 showed low toxicity in fibroblasts 24 h after treatment (10.1% death at
40 μ M), cytotoxicity was higher in HeLa and it increased with time (15.9% at 6 h vs.
23.5% at 24 h, at 40 μ M). Toxicity could therefore explain that the splicing modifier effect
of PK4C9 decreases with time in HeLa but not in fibroblasts. (Note that we use “% of
death at 40 μ M” instead of LD50 values, because cells did not reach >50 % death even
after curve saturation).

2. HeLa are cancer cells. Cancer cells are known to often overexpress transporters that
pump substances out of the cell, a mechanisms leading to drug resistance. This could
also explain why the splicing modifier effect of PK4C9 decreases with time in HeLa but

not in fibroblasts.

We have now **modified the text** to mention these points in the Fig 3a legend, which
reads as follows:

*“Note that PK4C9 and BJGF466 elicit maximal E7 inclusion at early time points, may be*
*due to progressive compound toxicity and/or to the molecules being secreted out of the*
*cell at later time points, as previously seen in cancer cells ().”*

**4. A time course (up to 72 hrs or longer) study in dose response format is needed**
**to make a confident statement about the cytotoxicity of these molecules. A 24 hr**
**cytotoxicity study as shown in Fig 3d is inadequate and a tad misleading although**
**it shows PK4C9 to be superior to the other tested molecules at one early timepoint**
**and at a given dose. Given that there are over 200 splicing events impacted by the**
**molecule a more thorough evaluation of cytotoxicity is warranted.**

For our previous version of the manuscript, 24 h cytotoxicity curves were obtained by the
LDH method for our screening hits NCI377363, PK4C9, BJGF466, and 288D. At this
time point, some of the molecules did not reach >50 % death, even after curve
saturation. Therefore their LD50 value could not be calculated. To be able to compare
the toxicity of these four hits, we represented the “% of death at 40 μ M” and “the % of
death at the concentration of maximum activity”, as shown in Fig. 3d. In our revised
version of the manuscript we have included our complete **cytotoxicity curves at 24 h**
(see **new Supp. Fig. S7**). In addition, and following the recommendation of the reviewer,
we have also expanded our cytotoxicity analysis to **72 h**. At this time point, the same
trend was observed, with two molecules still not reaching >50 % death, and PK4C9
being the least toxic of the four. 100 % death was not reached by any molecule at any
concentration.

**5a. In their RNA-Seq experiment the authors identified 290 transcripts with**
**modified splicing relative to DMSO. The scope of alternative splicing events**
**impacted by the compound may be an underestimate given that sequencing reads**
**could not unambiguously mapped on to full length transcripts. A more stringent**
**statistical assessment of the splicing changes and a rank ordering of the changes**
**based on significance would be very informative.**

To address the question raised by the reviewer, the following **exon junction analysis**
 was performed. Exon junctions with read support only in the treated but not in the control
 samples were analyzed. The majority (80.7%) of those junctions are not in
 RefSeq/Ensembl annotations (see **Table below**). However, more than half of the
 junctions are represented in the Intropolis database by both (55.4%) or one (39.3%) of
 the two junction coordinates. Thus, there is sequencing evidence for most of the
 junctions in already published RNASeq experiments, arguing against the concern that
 the scope of alternative splicing events impacted by PK4C9 may be an underestimate.
 Only 38 junctions (5.3%) detected in the treated but not in the control samples were not
 observed elsewhere.

Numbers of junctions represented in Intropolis and RefSeq/Ensembl GTF exon annotation		
	Intropolis Database	RefSeq/Ensembl GTF
begin and end coordinates present	399 (55.4%)	3 (0.4%)
only begin coordinate present	138 (19.2%)	62 (8.6%)
only end coordinate present	145 (20.1%)	74 (10.3%)
begin and end coordinate not present	38 (5.3%)	581 (80.7%)
Sum	720	720

This **new analysis and table** have now been incorporated to our revised Supporting
 Methods section.

**5b. Also, It would be good for the authors to clarify which of the splicing events**
 **are due to rescue of SMN2 splicing and which may be resulting from non-specific**
 **interactions of the compound with other RNA sequences or RNA secondary**
 **structures, genome wide. Comparison of the RNA Seq profile of PK4C9 in wild**
 **type versus SMA patient fibroblasts could offer insights on this front.**
 **Alternatively, comparison to SMN overexpression / rescue or to Spinraza**
 **treatment would be informative in this regard.**

Our RNA sequencing (RNA-seq) analysis detected 201 differentially spliced genes with
 an absolute PSI (percent spliced in) >0.4 upon treatment of human SMA fibroblasts
 (GM03813C) with PK4C9 (40 μ M, 24 h). A series of studies in mice have previously
 shown that reduction of SMN protein results in widespread splicing abnormalities, the
 identity of which depends on the genetic model, experimental conditions, and tissue/cell
 lines used. For example, the following numbers of dysregulated splicing events have
 been reported in SMN-depleted mice cells: 145 (motor neurons)¹²; 252 (spinal cord,

post-symptomatic stage), 16 (spinal cord, pre-symptomatic stage)¹³; 104 (motor
neurons), 86 (white matter)¹⁴; 259 (spinal cord), 73 (brain), and 633 (kidney)¹⁵. It is
therefore not surprising that the recovery of SMN protein induced by PK4C9 in SMA
fibroblasts is coupled with a large number of splicing changes, which could represent the
reversal of at least part of such generalized splicing abnormalities and be of therapeutic
relevance. ~25% of the changes found in our RNAseq study affect genes altered in
previous reports in mice. However, a formal comparison between ours and these
previous results has not been conducted in our study, given that the identity of specific
exons and introns affected in SMN-depleted mouse nerve cells has been shown to not
translate to human SMA fibroblasts¹².

Besides PK4C9, there are only two other examples of *SMN2*-splicing modifying small
molecules in the literature for which RNA-seq data also exist, the chemical scaffolds of
which differ notably from PK4C9. In particular:

(1) Novartis: NVS-SM1 (100 nM). 35 differentially spliced genes with PSI>0.4 were
identified¹⁰.

(2) Hoffmann-La Roche: SMN-C3 (500 nM). 13 differentially spliced genes with PSI>0.4
were identified⁹.

In these two cases, the molecules tested were not direct hits from a chemical screen
(like PK4C9), but chemically optimized leads with maximized cellular potency (nM range)
and oral availability. A fair comparison of these two molecules with PK4C9 can therefore
not be made, since PK4C9 is still in the pre-optimization stage. However, we did find
three differentially spliced genes (*SMN2*, *SLC25A17*, and *VPS29*) in common between
the three studies (see **Venn Diagram** below), further supporting that at least some of all
PK4C9-induced splicing changes represent a positive consequence of SMN protein
rescue.

Venn diagram. Genes where alternative splicing events were detected with an absolute PSI of at least 0.4 between treated and control samples. There are three genes that were affected by all three compounds. <http://bioinfogp.cnb.csic.es/tools/venny/index.html>. We would be happy to include this figure in the manuscript should the reviewer agree

However, we acknowledge that part of the PK4C9-sensitive splicing changes are also
likely to be off-targets. In this regard, it is important to keep in mind that PK4C9, in its
current state, is not intended as a therapeutic agent, but as a proof-of-concept molecule
that will undergo chemical optimization to become a more potent and specific lead
compound. Being able to discern between undesired PK4C9-induced off-target vs. SMN
recovery-mediated splicing changes is key for the chemical optimization of PK4C9's
specificity. In an initial low-scale attempt, we compared the effect of PK4C9 on eight of
these genes in SMA vs. WT fibroblasts. To do this, we assumed that (1) true off-targets
would be similarly affected by PK4C9 in WT and SMA cells, but that (2) SMN-dependent
changes would respond differently to treatment in WT vs. SMA cells, given their different
SMN starting levels (see Fig. 4d). Four out of these eight genes belonged to the first
case and the remaining four to the second, confirming the co-existence of both effects.
We now plan to expand this analysis to the rest of transcripts and to combine this
information with our structural results (see Fig. 5, Fig. 6 and Fig. 7), in order to lead the
optimization of PK4C9's specificity.

Finally, the less-active, **structural analogues of PK4C9** that do not affect *SMN2* E7
inclusion (see **new Supp. Fig. S8**) also failed to modify the splicing of two of the
transcripts that we classified as SMN-recovery dependent, further validating our
conclusions.

**Mentions to all these points** have now been added to the Results and Discussion
sections of the revised version of our manuscript.

**6. The authors should seriously consider including a structurally related, inactive**
**(in SMN assays) compound as a negative control in their key cellular and**
**biophysical studies, NGS etc**

As requested by the reviewer, **eight structural analogues** of PK4C9 have been
generated and their TSL2-binding and functional data added to the manuscript (see **new**
**Supp. Fig. S8**). Compared to PK4C9, all these analogues displayed reduced binding to
TSL2 in our fluorescence displacement assay. This reduction was coupled with a
proportional loss of *SMN2* E7 splicing modifier activity in HeLa cells (*SMN2*^{E6-7-8}
minigene) and SMA fibroblasts (endogenous *SMN2* transcript), which was in turn nicely
explained by their different binding poses found by molecular docking (see **new Supp.**

**Fig. S8**). Finally, the splicing of *LPIN1* and *RPS6KB1*, two transcripts found in our RNA-
seq analysis, the splicing changes of which were classified as SMN recovery-dependent,
were also not affected by the non-active PK4C9 analogues (see **new Supp. Fig. S8**).
Altogether, these results provide additional confirmation that the cellular activity of
PK4C9 is mediated by TSL2.

**7. In recent years there has been significant progress in identifying small**
**molecule and antisense-oligonucleotide based approaches to enhancing SMN2**
**splicing / exon 7 inclusion. Almost all of these approaches have relied on a couple**
**of mouse models of SMA to demonstrate in vivo efficacy. The current study**
**however provides evidence of in vivo efficacy in a less commonly used fly model**
**of SMA. This does not allow for proper benchmarking / comparison of TSL2**
**modulators to previously demonstrated approaches to enhancing SMN2 exon7**
**splicing that are currently in the clinic, which is critical given the current state of**
**the field.**

While we understand the point made by the reviewer, we would like to make the
following clarifications. We have also revised our manuscript text to clarify these points.

1. The *Drosophila* splicing sensor expresses a transgene with the human *SMN2* E6-7-8
sequence. PK4C9 was tested in this system to provide additional validation of its splicing
modifier activity, which in these flies can be easily and specifically targeted to motor
neurons. This validation was, however, not intended as a substitute of compound
evaluation in a mammalian SMA model, nor as proof of *in vivo* efficacy in the strict
sense. Indeed, the *Drosophila* splicing sensor cannot be considered an SMA disease
model, since these flies do not express pathogenic mutations.

2. Our study provides proof-of-concept to using small molecules to target TSL2 and
manipulate *SMN2* splicing, The hit molecules presented here are still in a pre-
optimization stage of development, and therefore we do not claim that they can be
considered as clinical candidates. However, we undertook an extensive effort to
describe and validate the mechanism of action of PK4C9, which would aid a subsequent
chemical optimization campaign for this molecule. It will be sensible that the subsequent
optimized lead compounds are assessed in the appropriate mice models, like the *Smn*

allele C⁵⁵ or the SMNdelta7⁵⁶. However, we strongly believe that testing PK4C9 in its
current state on SMA mice models would have been a non-justifiable use of resources
and animals, which given the animal experimentation legislation in Switzerland would
have most likely lead to a negative cost benefit analysis. Specifically, the SMNdelta7 is a
severe disease model, where animals of the relevant genotype have to be generated for
every experiment by genetic crossing. These animals are extremely ill and die within ~2
925 weeks from birth, making their generation to evaluate a non-optimized hit, for which we
also do not have PK or *in vivo* distribution data, unjustifiable.

**8. While Garcia-Lopez et al present promising, early evidence for the identification**
**of small molecule modulators of the TSL2 stem loop structure in SMN2 the study**
**fails to provide thorough validation and selectivity assessment of the**
**compound(s). The current study represents a modest, incremental increase in our**
**structural and mechanistic understanding of how TSL2 (which has long been**
**known to be a key regulatory region for SMN2 splicing) may be modulated with**
**small molecules to enhance SMN2 exon 7 inclusion. In summary, I would not**
**recommend accepting this manuscript for publication in its current form.**

We hope to have provided enough convincing data and arguments to address the
concerns of the reviewer, and to have presented the strengths of our study more clearly
in our revised version of the manuscript , as well as throughout this document.

- 1. Furtig, B., *et al.* Time-resolved NMR studies of RNA folding. *Biopolymers* **86**,
360-383 (2007).
- 2. Singh, N.N., Singh, R.N. & Androphy, E.J. Modulating role of RNA structure in
alternative splicing of a critical exon in the spinal muscular atrophy genes.
*Nucleic Acids Res* **35**, 371-389 (2007).
- 3. Burd, C.G. & Dreyfuss, G. RNA binding specificity of hnRNP A1: significance of
hnRNP A1 high-affinity binding sites in pre-mRNA splicing. *EMBO J* **13**, 1197-
1204 (1994).
- 4. Brichta, L., *et al.* Valproic acid increases the SMN2 protein level: a well-
known drug as a potential therapy for spinal muscular atrophy. *Human*
*molecular genetics* **12**, 2481-2489 (2003).
- 5. Hofmann, Y., Lorson, C.L., Stamm, S., Androphy, E.J. & Wirth, B. Htra2-beta 1
stimulates an exonic splicing enhancer and can restore full-length SMN
expression to survival motor neuron 2 (SMN2). *Proc Natl Acad Sci U S A* **97**,
9618-9623 (2000).
- 6. Cartegni, L., Hastings, M.L., Calarco, J.A., de Stanchina, E. & Krainer, A.R.
Determinants of exon 7 splicing in the spinal muscular atrophy genes, SMN1
and SMN2. *Am J Hum Genet* **78**, 63-77 (2006).
- 7. Zhang, M.L., Lorson, C.L., Androphy, E.J. & Zhou, J. An in vivo reporter system
for measuring increased inclusion of exon 7 in SMN2 mRNA: potential
therapy of SMA. *Gene therapy* **8**, 1532-1538 (2001).
- 8. Mayer, F., *et al.* Evolutionary conservation of vertebrate blood-brain barrier
chemoprotective mechanisms in *Drosophila*. *J Neurosci* **29**, 3538-3550
(2009).
- 9. Naryshkin, N.A., *et al.* Motor neuron disease. SMN2 splicing modifiers
improve motor function and longevity in mice with spinal muscular atrophy.
*Science* **345**, 688-693 (2014).
- 10. Palacino, J., *et al.* SMN2 splice modulators enhance U1-pre-mRNA association
and rescue SMA mice. *Nature chemical biology* **11**, 511-517 (2015).
- 11. Grzeschik, S.M., Ganta, M., Prior, T.W., Heavlin, W.D. & Wang, C.H.
Hydroxyurea enhances SMN2 gene expression in spinal muscular atrophy
cells. *Ann Neurol* **58**, 194-202 (2005).
- 12. Custer, S.K., *et al.* Altered mRNA Splicing in SMN-Depleted Motor Neuron-
Like Cells. *PloS one* **11**, e0163954 (2016).
- 13. Baumer, D., *et al.* Alternative splicing events are a late feature of pathology in
a mouse model of spinal muscular atrophy. *PLoS Genet* **5**, e1000773 (2009).
- 14. Zhang, Z., *et al.* Dysregulation of synaptogenesis genes antecedes motor
neuron pathology in spinal muscular atrophy. *Proc Natl Acad Sci U S A* **110**,
19348-19353 (2013).
- 15. Zhang, Z., *et al.* SMN deficiency causes tissue-specific perturbations in the
repertoire of snRNAs and widespread defects in splicing. *Cell* **133**, 585-600
(2008).
- 16. Underwood, J.G., *et al.* FragSeq: transcriptome-wide RNA structure probing
using high-throughput sequencing. *Nat Methods* **7**, 995-1001 (2010).

- 17. Siegfried, N.A., Busan, S., Rice, G.M., Nelson, J.A. & Weeks, K.M. RNA motif
discovery by SHAPE and mutational profiling (SHAPE-MaP). *Nat Methods* **11**,
959-965 (2014).
- 18. Zubradt, M., *et al.* DMS-MaPseq for genome-wide or targeted RNA structure
probing in vivo. *Nat Methods* **14**, 75-82 (2017).
- 19. Singh, N.N., Lee, B.M. & Singh, R.N. Splicing regulation in spinal muscular
atrophy by an RNA structure formed by long-distance interactions. *Annals of*
*the New York Academy of Sciences* **1341**, 176-187 (2015).
- 20. Kulik, M., Goral, A.M., Jasinski, M., Dominiak, P.M. & Trylska, J. Electrostatic
interactions in aminoglycoside-RNA complexes. *Biophysical journal* **108**, 655-
665 (2015).
- 21. Meiler, J. PROSHIFT: protein chemical shift prediction using artificial neural
networks. *J Biomol NMR* **26**, 25-37 (2003).
- 22. Han, B., Liu, Y., Ginzinger, S.W. & Wishart, D.S. SHIFTX2: significantly
improved protein chemical shift prediction. *J Biomol NMR* **50**, 43-57 (2011).
- 23. Xu, X.P. & Case, D.A. Automated prediction of ¹⁵N, ¹³Calpha, ¹³Cbeta and
¹³C' chemical shifts in proteins using a density functional database. *J Biomol*
*NMR* **21**, 321-333 (2001).
- 24. Kohlhoff, K.J., Robustelli, P., Cavalli, A., Salvatella, X. & Vendruscolo, M. Fast
and accurate predictions of protein NMR chemical shifts from interatomic
distances. *J Am Chem Soc* **131**, 13894-13895 (2009).
- 25. Atieh, Z., Aubert-Frecon, M. & Allouche, A.R. Rapid, accurate and simple
model to predict NMR chemical shifts for biological molecules. *J Phys Chem B*
**114**, 16388-16392 (2010).
- 26. Shen, Y. & Bax, A. SPARTA+: a modest improvement in empirical NMR
chemical shift prediction by means of an artificial neural network. *J Biomol*
*NMR* **48**, 13-22 (2010).
- 27. Cromsig, J.A., Hilbers, C.W. & Wijmenga, S.S. Prediction of proton chemical
shifts in RNA. Their use in structure refinement and validation. *J Biomol NMR*
**21**, 11-29 (2001).
- 28. Dejaegere, A., Bryce, R.A. & A., C.D. An Empirical Analysis of Proton Chemical
Shifts in Nucleic Acids. *ACS Symp. Ser.* **732**, 194-206 (1999).
- 29. Lange, O.F., van der Spoel, D. & de Groot, B.L. Scrutinizing molecular
mechanics force fields on the submicrosecond timescale with NMR data.
*Biophysical journal* **99**, 647-655 (2010).
- 30. Dalvit, C., *et al.* Identification of compounds with binding affinity to proteins
via magnetization transfer from bulk water. *J Biomol NMR* **18**, 65-68 (2000).
- 31. Muller, C.W., Schlauderer, G.J., Reinstein, J. & Schulz, G.E. Adenylate kinase
motions during catalysis: an energetic counterweight balancing substrate
binding. *Structure* **4**, 147-156 (1996).
- 32. Silvers, R., Keller, H., Schwalbe, H. & Hengesbach, M. Differential scanning
fluorimetry for monitoring RNA stability. *Chembiochem : a European journal*
*of chemical biology* **16**, 1109-1114 (2015).
- 33. Orry, A.J., Abagyan, R.A. & Cavasotto, C.N. Structure-based development of
target-specific compound libraries. *Drug discovery today* **11**, 261-266 (2006).

- 34. Harris, C.J., Hill, R.D., Sheppard, D.W., Slater, M.J. & Stouten, P.F. The design
and application of target-focused compound libraries. *Comb Chem High*
*Throughput Screen* **14**, 521-531 (2011).
- 35. Tran, T. & Disney, M.D. Identifying the preferred RNA motifs and chemotypes
that interact by probing millions of combinations. *Nature communications* **3**,
1125 (2012).
- 36. Thomas, J.R. & Hergenrother, P.J. Targeting RNA with small molecules. *Chem*
*Rev* **108**, 1171-1224 (2008).
- 37. Bodoor, K., *et al.* Design and implementation of an ribonucleic acid (RNA)
directed fragment library. *J Med Chem* **52**, 3753-3761 (2009).
- 38. Maly, D.J., Choong, I.C. & Ellman, J.A. Combinatorial target-guided ligand
assembly: identification of potent subtype-selective c-Src inhibitors. *Proc*
*Natl Acad Sci U S A* **97**, 2419-2424 (2000).
- 39. Kralovicova, J., Patel, A., Searle, M. & Vorechovsky, I. The role of short RNA
loops in recognition of a single-hairpin exon derived from a mammalian-wide
interspersed repeat. *RNA Biol* **12**, 54-69 (2015).
- 40. Disney, M.D., Yildirim, I. & Childs-Disney, J.L. Methods to enable the design of
bioactive small molecules targeting RNA. *Org Biomol Chem* **12**, 1029-1039
(2014).
- 41. Guan, L. & Disney, M.D. Recent advances in developing small molecules
targeting RNA. *ACS chemical biology* **7**, 73-86 (2012).
- 42. Disney, M.D. Rational design of chemical genetic probes of RNA function and
lead therapeutics targeting repeating transcripts. *Drug discovery today*
(2013).
- 43. Gareiss, P.C., *et al.* Dynamic combinatorial selection of molecules capable of
inhibiting the (CUG) repeat RNA-MBNL1 interaction in vitro: discovery of
lead compounds targeting myotonic dystrophy (DM1). *J Am Chem Soc* **130**,
16254-16261 (2008).
- 44. Warf, M.B., Nakamori, M., Matthys, C.M., Thornton, C.A. & Berglund, J.A.
Pentamidine reverses the splicing defects associated with myotonic
dystrophy. *Proc Natl Acad Sci U S A* **106**, 18551-18556 (2009).
- 45. Velagapudi, S.P., *et al.* Design of a small molecule against an oncogenic
noncoding RNA. *Proc Natl Acad Sci U S A* **113**, 5898-5903 (2016).
- 46. Stelzer, A.C., *et al.* Discovery of selective bioactive small molecules by
targeting an RNA dynamic ensemble. *Nature chemical biology* **7**, 553-559
(2011).
- 47. Patwardhan, N.N., *et al.* Amiloride as a new RNA-binding scaffold with
activity against HIV-1 TAR. *Medchemcomm* **8**, 1022-1036 (2017).
- 48. Velagapudi, S.P. & Disney, M.D. Two-dimensional combinatorial screening
enables the bottom-up design of a microRNA-10b inhibitor. *Chem Commun*
*(Camb)* **50**, 3027-3029 (2014).
- 49. Di Giorgio, A., Tran, T.P. & Duca, M. Small-molecule approaches toward the
targeting of oncogenic miRNAs: roadmap for the discovery of RNA
modulators. *Future Med Chem* **8**, 803-816 (2016).

- 50. Luo, Y. & Disney, M.D. Bottom-up design of small molecules that stimulate
exon 10 skipping in mutant MAPT pre-mRNA. *Chembiochem : a European*
*journal of chemical biology* **15**, 2041-2044 (2014).
- 51. Liu, Y., Rodriguez, L. & Wolfe, M.S. Template-directed synthesis of a small
molecule-antisense conjugate targeting an mRNA structure. *Bioorg Chem* **54**,
7-11 (2014).
- 52. Reichert, C., Perozzo, R. & Borchard, G. Non-covalent modification of
granulocyte-colony stimulating factor (G-CSF) by coiled-coil technology. *Int J*
*Pharm* **511**, 98-103 (2016).
- 53. Perozzo, R., Folkers, G. & Scapozza, L. Thermodynamics of protein-ligand
interactions: history, presence, and future aspects. *J Recept Signal Transduct*
*Res* **24**, 1-52 (2004).
- 54. Seeliger, M.A., *et al.* c-Src binds to the cancer drug imatinib with an inactive
Abl/c-Kit conformation and a distributed thermodynamic penalty. *Structure*
**15**, 299-311 (2007).
- 55. Osborne, M., *et al.* Characterization of behavioral and neuromuscular junction
phenotypes in a novel allelic series of SMA mouse models. *Human molecular*
*genetics* **21**, 4431-4447 (2012).
- 56. Le, T.T., *et al.* SMNDelta7, the major product of the centromeric survival
motor neuron (SMN2) gene, extends survival in mice with spinal muscular
atrophy and associates with full-length SMN. *Human molecular genetics* **14**,
845-857 (2005).

Reviewers' Comments:

Reviewer #1:

Remarks to the Author:

The authors have replied to most of my comments satisfactorily.

However I would like to make a couple of comments regarding the replies (numbering as in the original review):

1- It would have been nice that the RT-PCR gel consistent with the bar graph was already there in the first version.

6- My experience with these techniques contradicts partially the author's statement: "qPCR has higher sensitivity and reliability over a greater dynamic range of RNA concentrations than other techniques, including semi-quantitative PCR or Northern blot." While there is no doubt about higher sensitivity of PCR methods, Northern blots are far more reliable than qPCR because, not depending on amplification methods, are more representative of the real mRNA concentrations in the cell/tissue. The fact that the authors do not have access to radioactive nucleotides is understandable but does not justify such sweeping statement.

Reviewer #2:

Remarks to the Author:

The new experiments and discussion clearly add to the contribution of this paper, and all of my concerns from my previous review have been addressed. I recommend an accept.

Reviewer #3:

Remarks to the Author:

No further revisions are required. Proposed changes & explanation were sufficient and the manuscript can be accepted.

Reviewer #5:

Remarks to the Author:

The concerns raised by all the reviewers were thoroughly addressed by the authors. Additional experiments were performed and corrections to the manuscript were made. Regarding the issues raised by Reviewer 4:

1) The authors provide sufficient evidence for the significance of the study. Specifically, the influence of triloop and pentaloop conformations of TSL2 on E7 splicing and the first NMR structure of TSL2. These contributions can aid in the further development of small molecules in the treatment of SMA and can be extrapolated to other splicing-mediated diseases targeting RNA.

2) Regarding the selectivity of binding of PK4C9 to TSL2 the authors demonstrate that the compound does not bind to unrelated RNA used in their study (Supple Fig S6).

3) The concern regarding the metal ion contaminants causing a change in the TSL2 conformation were appropriately addressed by carrying out elemental analysis of PK4C9 sample, which showed presence of metals (Pd, Mg, Fe) in trace quantities. Also, the use of 0.1 mM EDTA as a metal chelator in the NMR buffer ruled out the possibility of conformational change attributed to the presence of metal ions.

4) The shift in the compound spectra on binding to TSL2 was demonstrated by WaterLOGSY experiment.

5) An attempt to conduct direct binding assays was done by using SPR and ITC assays. However, due to weak binding affinity and solubility issues, these experiments were not conducted at higher

concentrations to determine the K_d value. In addition, the authors conducted fluorescent displacement assay with Ribogreen dye to rule out a false positive obtained by using TO-PRO1 dye.

6) Regarding the dose responsiveness of PK4C9, the text was appropriately modified in the legend for Fig 3.

7) Similarly, the maximal E7 inclusion at early time point observed in case of PK4C9 and BJGF466 in HeLa cells was attributed to higher toxicity of these compounds in HeLa cells lines compared to fibroblasts and presence of transporters in HeLa cells that pump out compounds from the cells. The text was modified in the legend for Fig 3a.

8) Cytotoxicity analysis up to 72 hours was performed as suggested by the reviewer.

9) To confirm that the activity of PK4C9 was mediated by binding to TSL2, eight additional analogs of PK4C9 were tested to correlate the TSL2 binding with the % E7 inclusion.

10) The authors suggested that the studies with PK4C9 were proof-of-concept studies and the molecule was not intended to be used as a therapeutic compound. Due to this reason, the authors did not test PK4C9 in mice models of the disease.

11) I find it most concerning that a K_d of 150-200- μ M is found for the compound and yet the cellular activity is 10-fold lower concentration. Does this not argue for activity being due to non-specific effects. this must be addressed

We thank all Reviewers for their time and positive feedback. We include responses to the minor points of Reviewer 1 and 5 below. Changes in the manuscript have been highlighted in yellow. We trust that the Reviewers will now find our manuscript suitable for publication, as already agreed by Reviewer 2 and 3.

Reviewer #1

The authors have replied to most of my comments satisfactorily. However I would like to make a couple of comments regarding the replies (numbering as in the original review):

1) It would have been nice that the RT-PCR gel consistent with the bar graph was already there in the first version.

We agree with the Reviewer and we apologize for not having shown a better image in our first version of the manuscript to represent the 8 replicates (3 biological and 2-3 technical) quantified in Fig 1c.

6) My experience with these techniques contradicts partially the author's statement: "qPCR has higher sensitivity and reliability over a greater dynamic range of RNA concentrations than other techniques, including semi-quantitative PCR or Northern blot." While there is no doubt about higher sensitivity of PCR methods, Northern blots are far more reliable than qPCR because, not depending on amplification methods, are more representative of the real mRNA concentrations in the cell/tissue. The fact that the authors do not have access to radioactive nucleotides is understandable but does not justify such sweeping statement.

We apologize for our unfortunate wording. It was not our intention to question the reliability of Northern blot, as we are aware of the advantages of this technique, including the direct visualization of RNA bands. Our statement referred to the potentially wider range of RNA concentrations that can be quantified by qPCR. The following **text** has now been included to the Materials and Methods section of our manuscript (**page 18**) to complement the point raised by the Reviewer: *"qPCR. [...]. Due to the low copy number of SMN2 transcripts, non-radioactive Northern blot failed to show visible SMN2 bands and could not be used as a validation technique. SMN2 isoform bands were shown by semi-quantitative PCR instead"*.

Reviewer #5:

The concerns raised by all the reviewers were thoroughly addressed by the authors. Additional experiments were performed and corrections to the manuscript were made. Regarding the issues raised by Reviewer 4

[Points 1-10 removed as these were addressed]

11) I find it most concerning that a Kd of 150-200- uM is found for the compound and yet the cellular activity is 10-fold lower concentration. Does this not argue for activity being due to non-specific effects. this must be addressed.

First, we thank Reviewer #5 for their time assessing our response to Reviewer's #4 comments and for acknowledging our efforts in this regard. With respect to comment 11, we find it unlikely that activity is due to non-specific effects, as there are other more plausible explanations for the mismatch between the predicted Kd and cellular activity, detailed in points 1-3 below.

(1) We have shown that the effect of PK4C9 on SMN2 E7 inclusion depends on the

structural integrity of TSL2 (Fig. 5 b, c). Moreover, we were able to link the effect of PK4C9 on E7 inclusion to TSL2 binding efficiency by studying eight structural analogues of this compound (Supp. Fig. S8). These observations demonstrate that the splicing modifier activity of PK4C9 is mediated by TSL2 and argue against an off-target as the responsible for the observed *SMN2* splicing changes.

(2) The EC₅₀ value of PK4C9 in our splicing cellular assay is ~25 μ M. Consistently, the EC₅₀ value of PK4C9 in our TO-PRO-1 binding assay is 16 μ M (Fig. 2c). Despite our efforts, ITC failed to provide an experimentally calculated K_d value for PK4C9, and an estimated K_d of 100-200 μ M (which is ~3.5-to-7 times higher than the cellular EC₅₀) was proposed. However, this K_d value may be an underestimate, given the poor solubility of PK4C9 in ITC buffer, which could lead to invisible precipitation of the compound and a reduction of its real concentration in solution. It is likely that the solubility of PK4C9 in TO-PRO-1 assay buffer and cell culture medium is better than in ITC buffer, which would increase the percentage of PK4C9 molecules available to reach TSL2.

(3) The ratio between folded and unfolded TSL2 states likely differs in vitro and in vivo due to molecular-crowding effects. Molecular crowding tends to favor the folded state of RNA inside the cell compared to in vitro conditions, which could influence target recognition¹. Moreover, our cellular and ITC assays measure different things (namely, splicing and binding, respectively). In cells, the extent and lifetime of the conformational effect of PK4C9 on TSL2 may be sufficient to promote the downstream splicing effect, despite the low predicted K_d of the compound in vitro².

Because of its estimated nature, and given the above points, the K_d value proposed by ITC was given in our answer to Reviewer #4 but was not included in the main text of our manuscript. However, in reference to the point raised by Reviewer #5, the following **text** has now been included to the Results section of our manuscript (**page 7**): “(PK4C9) showed the strongest effect, with an average E7 inclusion of up to 72% at 40 μ M (43% increase respect to DMSO-treated cells) corresponding to an EC₅₀ value of ~25 μ M that is consistent with the EC₅₀ value of PK4C9 in the TO-PRO-1 binding assay (16 μ M, Fig. 2c)”. We hope that our explanations and text modification have addressed the concern of the Reviewer satisfactorily.

1. Dupuis, N.F., Holmstrom, E.D. & Nesbitt, D.J. Molecular-crowding effects on single-molecule RNA folding/unfolding thermodynamics and kinetics. *Proc Natl Acad Sci U S A* **111**, 8464-8469 (2014).
2. Copeland, R.A., Pompliano, D.L. & Meek, T.D. Drug-target residence time and its implications for lead optimization. *Nature reviews. Drug discovery* **5**, 730-739 (2006).